# GTMGC: Using Graph Transformer to Predict Molecule's Ground-State Conformation

**Guikun Xu[1], Yongquan Jiang[1, 2, 3, ✉], Pengchuan Lei[1], Yan Yang[1, 2, 3], Jim X. Chen[4]**

[1]School of Computing and Aritifical Intelligence, Southwest Jiaotong University, Chengdu, China
[2]Institute of Aritifical Intelligence, Southwest Jiaotong University, Chengdu, China
[3]Engineering Research Center of Sustainable Urban Intelligent Transportation, Ministry of Education, China
[4]Department of Computer Science, George Mason University, Fairfax, VA, USA
`richxu945@gmail.com;{yqjiang@,yyang@,mylpc@my.}swjtu.edu.cn;jimxchen@gmail.com`

## ABSTRACT

The ground-state conformation of a molecule is often decisive for its properties. However, experimental or computational methods, such as density functional theory (DFT), are time-consuming and labor-intensive for obtaining this conformation. Deep learning (DL) based molecular representation learning (MRL) has made significant advancements in molecular modeling and has achieved remarkable results in various tasks. Consequently, it has emerged as a promising approach for directly predicting the ground-state conformation of molecules. In this regard, we introduce GTMGC, a novel network based on Graph-Transformer (GT) that seamlessly predicts the spatial configuration of molecules in a 3D space from their 2D topological architecture in an end-to-end manner. Moreover, we propose a novel self-attention mechanism called Molecule Structural Residual Self-Attention (MSRSA) for molecular structure modeling. This mechanism not only guarantees high model performance and easy implementation but also lends itself well to other molecular modeling tasks. Our method has been evaluated on the Molecule3D benchmark dataset and the QM9 dataset. Experimental results demonstrate that our approach achieves remarkable performance and outperforms current state-of-the-art methods as well as the widely used open-source software RDkit.

## 1 INTRODUCTION

The molecular ground-state conformation refers to the lowest energy state of a molecule on its potential energy surface. It represents the most stable configuration, requiring the least amount of energy to maintain. Additionally, it plays a crucial role in determining the physical, chemical, and biological properties of the molecule in most cases.

Recently, MRL methods based on DL have made significant strides in the field of molecular modeling (Wigh et al., 2022). This approach represents molecules as graphs, where atoms and bonds are respectively depicted as nodes and edges with features. By utilizing graph neural networks (GNNs) (Kipf & Welling, 2016; Veličković et al., 2017; Li et al., 2020; Ying et al., 2021; Kim et al., 2022) to learn the representation of molecules, a variety of tasks such as molecular property prediction, molecular generation, and molecular optimization can be accomplished (Gilmer et al., 2017; Shi et al., 2020; Luo et al., 2022; Zhang et al., 2023b). Of particular note, some studies (Wu et al., 2018; Townshend et al., 2020; Liu et al., 2021b; Luo et al., 2022; Liao & Smidt, 2022) have attempted to incorporate the ground-state 3D geometric information of molecules, such as atomic coordinates and interatomic distance matrix, as additional inputs to the neural networks. Such rich and advantageous inductive bias has significantly enhanced the model's performance across a wide range of tasks.

It is clear that studying molecular ground-state conformation is of great significance. However, obtaining the conformations of molecules is a challenging task. At present, the primary methods include molecular dynamics simulations (De Vivo et al., 2016), rough approximations through manually hand-designed force fields (Rappé et al., 1992; Halgren, 1996), or computationally intensive (DFT) calculations (Parr et al., 1979) and so on. These methods either demand substantial experimental and computational resources or are too imprecise. Therefore, finding an efficient way to obtain molecular ground-state conformation is an important research topic. Due to the success

of DL methods in the field of molecular modeling and the success of AlphaFold2 (Jumper et al., 2021) in protein structure prediction, using deep neural networks (DNNs) to directly predict them is a highly promising approach.

**Related Work.**  In previous studies (Mansimov et al., 2019; Simm & Hernández-Lobato, 2019; Xu et al., 2021b;c; Wu et al., 2022; Morehead & Cheng, 2023), deep generative models have been employed to generate low-energy stable conformations of molecules. However, these generative methods mainly focused on producing a multitude of potential stable conformations, rather than specifically targeting the ground-state conformation. As a result, additional screening steps were required in real-world applications to identify the best conformation among the generated ensemble. Recently, a novel benchmark called Molecule3D (Xu et al., 2021d) has emerged as a valuable resource for predicting the ground-state geometry of molecules. This benchmark comprises an extensive dataset consisting of nearly 4 million molecules along with their corresponding ground-state conformations. In Sec. A, more detailed related work has been organized.

Building upon the Molecule3D benchmark, we propose a GT-based (Dwivedi & Bresson, 2020; Ying et al., 2021; Kim et al., 2022; Min et al., 2022) model for predicting molecule's ground-state conformation. Our model is capable of seamlessly transforming molecular 2D graph into its stable 3D ground-state conformation through end-to-end prediction, eliminating the need for additional intermediate variables such as predicting the interatomic distance matrix first and then recovering the 3D coordinates from it (Simm & Hernández-Lobato, 2019; Shi et al., 2021).

**Implementation.**  In detail, The hole overview of our model's architecture is shown in Fig. 2. Firstly, the newly introduced context-aware Mole-BERT Tokenizer (Xia et al., 2023) is creatively employed to categorize identical atoms into distinct subclasses, mitigating the quantitative divergence between prevalent and scarce atoms. Each atom of molecules is tokenized into discrete values with chemical significance (e.g., carbon to carbon in benzene ring or aldehyde.), serving as Input IDs. Then, the widely used Laplacian Positional Encoding (Dwivedi & Bresson, 2020) in the graph domain which can be solely computed from graph's adjacency matrix without any complex or excessive prior knowledge is applied to encode the positional information of each node within the entire graph. Furthermore, A novel self-attention mechanism, termed Molecule Structural Residual Self-Attention (MSRSA), has been proposed. This mechanism utilizes commonly available and computationally inexpensive molecular structural information, adjacency matrix and creative *row-subtracted interatomic distance matrix* to compute residual bias terms upon self-attention scores ($\mathbf{QK}^T$), to enhance the self-attention mechanism's ability to model molecular structures. It largely retains the structure and elegance of the original Transformer (Vaswani et al., 2017) network, unlike other works (Ying et al., 2021; Li et al., 2022) that necessitate complex design and computation. Finally, we use a feed-forward network (FFN) head on atoms' representations to predict the molecular 3D ground-state conformation, achieving end-to-end prediction from the molecule's 2D graph.

**Results and Contribution.**  ($i$). Our approach, GTMGC, that predicts molecular 3D ground-state conformation from its 2D graph topological representation in an end-to-end manner, has been validated on the recently proposed benchmark dataset Molecule3D (Xu et al., 2021d) and the widely used QM9 (Ramakrishnan et al., 2014) dataset. Experimental results demonstrate that our approach outperforms the current best method (Xu et al., 2021d) and surpasses the DG (Havel, 1998) and ETKDG (Riniker & Landrum, 2015) algorithms implemented in the most commonly used open-source software RDkit (Landrum et al., 2013). ($ii$). GTMGC employs a distinctive input form. We innovatively leverage the MoleBERT Tokenizer (Xia et al., 2023) to discretize each atom of the molecule into chemically significant tokens, serving as the model's input, rather than conventional atom type IDs or Ogb-style embeddings (Hu et al., 2021). This effectively helps the model predict molecule's ground-state conformation. ($iii$). Furthermore, ablation studies strongly demonstrate the effectiveness of our method. ($iv$). More experiments in Sec. B show that the proposed MSRSA module can be effectively applied to many other molecular property prediction tasks. ($v$). The source code of our method is available at `https://github.com/Rich-XGK/GTMGC`.

## 2 PRELIMINARY

### 2.1 NOTATION AND PROBLEM DEFINITION

**Notation.**  For a molecule with $n$ atoms and $m$ bonds, its 2D structure representation is denoted as $\mathcal{G} = \{\mathcal{V}, \mathcal{E}\}$, where $\mathcal{V} = \{v_1, v_2, \ldots, v_n\}$ represents the set of $n$ atoms and $\mathcal{E} = \{e_1, e_2, \ldots, e_m\}$ represents the set of $m$ bonds. Each atom $v_i$ and bond $e_j$ may possess its own features, such as

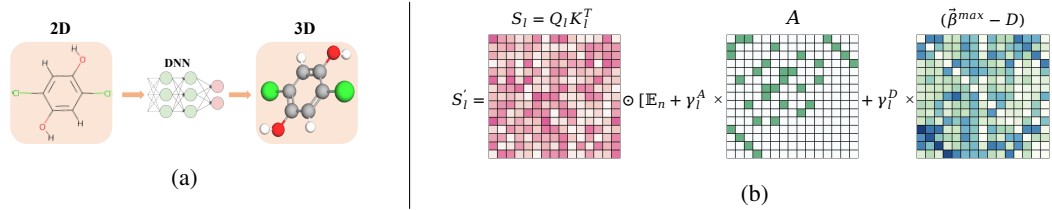

Figure 1: **(a)**: The overview of the ground-state conformation prediction task; **(b)**: Computation process of the attention map in MSRSA module, viewed through the lens of attention mask.

atom type, bond type, etc. In the context of a graph structure $\mathcal{G}$, its adjacency matrix is denoted by $\mathbf{A} \in \mathbb{R}^{n \times n}$, where $\mathbf{A}_{ij} = 1$ signifies the presence of an edge connecting node $v_j$ to $v_i$, while $\mathbf{A}_{ij} = 0$ indicates the absence of such an edge. The ground-state conformation of a molecule is denoted as $\mathbf{G} \in \mathbb{R}^{n \times 3}$, which represents a set of 3D Cartesian coordinates. Expanding on this, the interatomic distance matrix of a molecule is defined as $\mathbf{D} \in \mathbb{R}^{n \times n}$, where $\mathbf{D}_{ij}$ represents the Euclidean distance between atom $v_j$ and $v_i$.

**Problem Definition.** As depicted in Fig. 1 (a), the ground-state conformation prediction task aims to predict the conformation of a molecule in its ground state, $\mathbf{G} \in \mathbb{R}^{n \times 3}$, solely based on its 2D molecular structure, $\mathcal{G} = \{\mathcal{V}, \mathcal{E}\}$.

## 2.2 TRANSFORMER

In vanilla Transformer (Vaswani et al., 2017), the multi-head scaled dot-product self-attention (MHSA) module is solely responsible for modeling relationships between different elements. By calculating the similarity between queries and keys, it determines the importance of different elements to a single element. The weights are then normalized and used to compute a weighted sum of the values, resulting in a new representation for the corresponding element. The use of multiple heads allows different attention heads to learn distinct information within separate subspaces.

**Self-Attention.** Suppose the input fed into the MHSA module is $\mathbf{X} \in \mathbb{R}^{n \times d_{model}}$, where $n$ represents the number of tokens and $d_{model}$ represents the hidden dimension of the model. $\mathbf{W}^Q \in \mathbb{R}^{d_{model} \times d_k}$, $\mathbf{W}^K \in \mathbb{R}^{d_{model} \times d_k}$, and $\mathbf{W}^V \in \mathbb{R}^{d_{model} \times d_v}$ are three learnable linear mappings that map $\mathbf{X}$ to $\mathbf{Q} \in \mathbb{R}^{n \times d_k}$, $\mathbf{K} \in \mathbb{R}^{n \times d_k}$ and $\mathbf{V} \in \mathbb{R}^{n \times d_v}$, respectively. The scaled dot-product attention is then computed as:

$$\mathbf{Q} = \mathbf{X}\mathbf{W}^Q, \mathbf{K} = \mathbf{X}\mathbf{W}^K, \mathbf{V} = \mathbf{X}\mathbf{W}^V \tag{1}$$

$$\text{Attention}(\mathbf{Q}, \mathbf{K}, \mathbf{V}) = \text{softmax}(\frac{\mathbf{Q}\mathbf{K}^T}{\sqrt{d_k}})\mathbf{V}, \tag{2}$$

**Multi-Head.** After introducing h heads, total of h $\mathbf{W}_l^{\mathbf{Q}} \in \mathbb{R}^{d_{model} \times \frac{d_k}{n}}$, $\mathbf{W}_l^K \in \mathbb{R}^{d_{model} \times \frac{d_k}{h}}$ and $\mathbf{W}_l^V \in \mathbb{R}^{d_{model} \times \frac{d_v}{h}}$, map $\mathbf{X}$ to $\mathbf{Q}_l \in \mathbb{R}^{n \times \frac{d_k}{h}}$, $\mathbf{K}_l \in \mathbb{R}^{n \times \frac{d_k}{h}}$ and $\mathbf{V}_l \in \mathbb{R}^{n \times \frac{d_v}{h}}$, where $l \in [1, h]$. Then one head of scaled dot-product attention can be computed as equation 3. Let $\mathbf{W}^O \in \mathbb{R}^{d_v \times d_{model}}$ be another learnable linear mapping to map the concatenated output of all heads to the output of the MHSA module:

$$\mathbf{O}_l = \text{Attention}(\mathbf{Q}_l, \mathbf{K}_l, \mathbf{V}_l) = \text{softmax}(\frac{\mathbf{Q}_l \mathbf{K}_l^T}{\sqrt{d_k}})\mathbf{V}_l \tag{3}$$

$$\text{MHSA}(\mathbf{X}) = \text{Concat}(\mathbf{O}_1, \mathbf{O}_2, \dots, \mathbf{O}_h)\mathbf{W}^O \tag{4}$$

**Transformer Layer.** After MHSA, a feed-forward network (FFN) consisting of two linear transformations with a non-linear activation is applied to each element respectively and identically. Both the output of MHSA and FFN are first passed through a residual connection (He et al., 2016) and then normalized by a LayerNorm (Ba et al., 2016). The output of a Transformer layer can be computed as:

$$\mathbf{X}' = \text{LayerNorm}(\mathbf{X} + \text{MHSA}(\mathbf{X})) \tag{5}$$

$$\text{Layer}(\mathbf{X}) = \text{LayerNorm}(\mathbf{X}' + \text{FFN}(\mathbf{X}')) \tag{6}$$

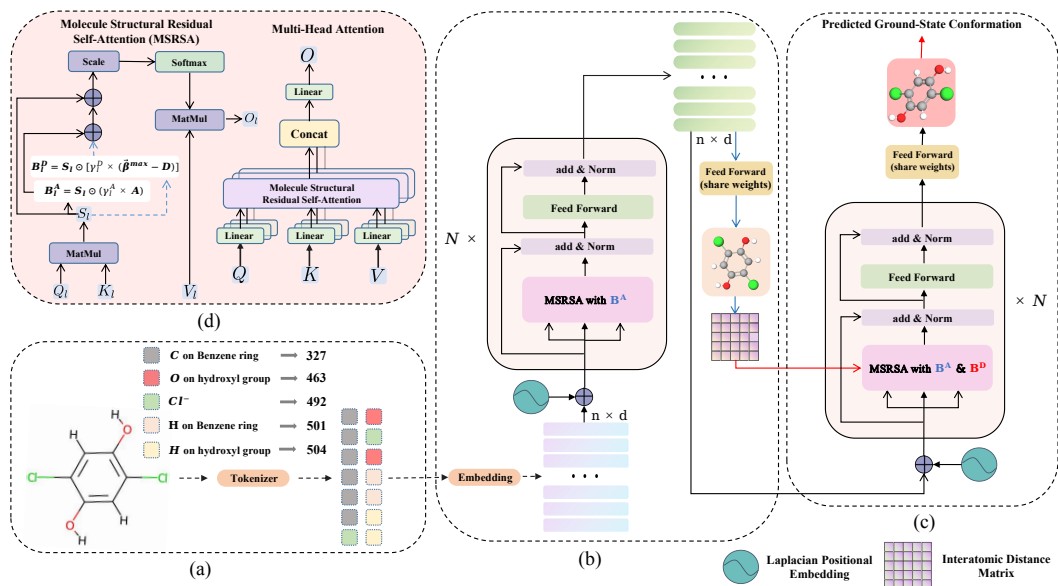

Figure 2: Overview of GTMGC. **(a)**: Atoms in the molecule are tokenized by MoleBERT Tokenizer (Xia et al., 2023) into chemically meaningful discrete codes as model's Input IDs. **(b)**: The encoder of GTMGC consisting of $N$ MSRSA blocks with only adjacency matrix residual bias term $\mathbf{B}^A$. **(c)**: The Decoder of GTMGC consisting of MSRSA blocks with both adjacency matrix residual bias term $\mathbf{B}^A$ and interatomic distance matrix residual bias term $\mathbf{B}^D$. **(d)**: The main Molecule Structural Residual Self-Attention (MSRSA) module, using residual bias terms $\mathbf{B}^A$ and $\mathbf{B}^D$ to help pure dot-product self-attention mechanism model molecule's structure information.

## 3 METHOD

### 3.1 MODEL INPUT

**Input IDs.** Thanks to the convenience of (Xia et al., 2023), we can now use the chemically meaningful IDs tokenized by the MoleBERT Tokenizer as input IDs for our model, as shown in Fig. 2(a). Since these tokens are assumed to already contain ample chemical information about atoms, we have ceased utilizing edge features to enhance the representation of molecular input features, thereby making the model more concise.

**Positional Encoding.** For graph structures, incorporating positional encoding becomes even more vital and challenging since elements in graph structures lack fixed positions. It is necessary to differentiate the positions of various nodes within the graph structure, allowing the model to capture the relationships between nodes. To adhere to the principle of introducing minimal prior knowledge to the model and enhancing its generality, we opted for the Laplacian Positional Encoding as (Dwivedi & Bresson, 2020) does, among using various graph structure positional encoding techniques (Zhang et al., 2020; Ying et al., 2021; Hussain et al., 2021; Park et al., 2022). This approach encodes positional relationships between nodes using the eigenvectors of the graph Laplacian matrix. It can be solely computed from every molecular graph's adjacency matrix without any complex or excessive prior knowledge.

Thus, our final input, $\mathbb{H} \in \mathbb{R}^{n \times d_{model}}$, is solely obtained by adding the Laplacian Positional Encoding vectors, $\mathbf{L} \in \mathbb{R}^{n \times n}$, to the feature vectors, $\mathbf{E} \in \mathbb{R}^{n \times d_{model}}$, embedded from the MoleBERT-tokenized input IDs.

### 3.2 GRAPH STRUCTURAL RESIDUAL SELF-ATTENTION (MSRSA)

In Section 2.2, we review in detail how a vanilla Transformer layer performs forward calculations. Its original dot-product self-attention mechanism models the relationships between all nodes in an undirected fully connected graph, similar to a GNN model (Kipf & Welling, 2016; Veličković et al., 2017; Gilmer et al., 2017) that performs message passing on an undirected fully connected graph.

For molecules, the local structure of atoms (such as functional groups) and the connectivity of edges (whether they have chemical bonds or not) have a huge impact on their chemical and structural properties. Since the original self-attention mechanism is unable to effectively capture these information, we propose an extended self-attention mechanism based on molecular structure, named Molecule

Structural Residual Self-Attention (MSRSA), as can be observed in Fig. 2(d), which incorporates the adjacency matrix, $\mathbf{A} \in \mathbb{R}^{n \times n}$, and interatomic distance matrix, $\mathbf{D} \in \mathbb{R}^{n \times n}$, of the molecule as residual bias terms for self-attention scores, $\mathbf{S}_l = \mathbf{Q}_l \mathbf{K}_l^T \in \mathbb{R}^{n \times n}$.

**Global.** Unlike the approach used by (Dwivedi & Bresson, 2020) that employs the adjacency matrix as a full attention mask to restrict the current node's focus to neighboring nodes, our method is based on retaining complete global attention. This brings our model closer to the original Transformer, as we assume that the original self-attention mechanism captures global information by focusing on all atoms in the molecule. Therefore, the basic self-attention scores calculated by full self-attention in each head $l \in [1, h]$ are:

$$\mathbf{S}_l = \mathbf{X}\mathbf{W}_l^Q (\mathbf{X}\mathbf{W}_l^K)^T = \mathbf{Q}_l \mathbf{K}_l^T \tag{7}$$

**Nearby.** To capture something useful about the molecule's local structure, the adjacency matrix $\mathbf{A}$ has been introduced as a residual bias term $\mathbf{B}_l^A \in \mathbb{R}^{n \times n}$ to $\mathbf{S}_l$, as shown in Fig. 2(d). $\mathbf{B}_l^A$ models atom's *1-hop* local structure information by attending to neighboring atoms connected to the current atom. For each head, we first multiply the adjacency matrix $\mathbf{A}$ by a learnable parameter $\gamma_l^A$ to automatically learn the requirement for local information. The result is then used in a hadamard product with $\mathbf{S}_l$ to obtain $\mathbf{B}_l^A$:

$$\mathbf{B}_l^A = \mathbf{S}_l \odot (\gamma_l^A \times \mathbf{A}) \tag{8}$$

**Spatial.** To enable the model to represent the spatial structure of molecules, introducing the interatomic distance matrix of molecules is a direct and simple approach. Similarly, we use the interatomic distance matrix of molecules as another residual bias term $\mathbf{B}_l^D \in \mathbb{R}^{n \times n}$, as shown in Fig. 2(d). $\mathbf{B}_l^D$ models molecule's spatial structure by assigning more weight to atoms closer to the current atom, which introduces the simple *distance relevance assumption*: the greater the distance between atomic elements, the lower the interaction (Choukroun & Wolf, 2021). For each head, we first subtract the row-max value, $\beta_i^{\max}, i \in [1, n]$ which represents the furthest distance from other nodes to the current node $v_i$, from each row of $\mathbf{D}$ to obtain **row-subtracted distance matrix** $\mathbf{D}_{\text{row-sub}}$, representing the *distance relevance assumption*. It is worth noting that, as far as we know, this represents our biggest difference from others who use the interatomic distance matrix to introduce molecule's spatial structure information and its effectiveness is supported by experimental evidence (Table 3; Table 4). Then another learnable parameter $\gamma_l^D$ is multiplied to $\mathbf{D}_{\text{row-sub}}$ to automatically ascertain the necessity for spatial information. The result is finally used in a hadamard product with $\mathbf{S}_l'$ to obtain $\mathbf{B}_l^D$:

$$\mathbf{D}_{\text{row-sub}} = \vec{\beta}^{\max} - \mathbf{D}, \; \vec{\beta}^{\max} = \max_{i \in [1,n]} \mathbf{D} \tag{9}$$

$$\mathbf{B}_l^D = \mathbf{S}_l \odot (\gamma_l^D \times \mathbf{D}_{\text{row-sub}}) \tag{10}$$

Thereafter, we sum up $\mathbf{S}_l$, $\mathbf{B}_l^A$ and $\mathbf{B}_l^D$ together, where $\mathbf{B}_l^A$ and $\mathbf{B}_l^D$ are residual bias terms of $\mathbf{S}_l$, and then scale the results by $\sqrt{d_k}$, and finally normalize it using a softmax function to obtain the final output $\mathbf{O}_l$ of each head:

$$
\begin{aligned}
\mathbf{S}_l' &= (\mathbf{S}_l + \mathbf{B}_l^A + \mathbf{B}_l^D) \\
&= \mathbf{S}_l + \mathbf{S}_l \odot (\gamma_l^A \times \mathbf{A}) + \mathbf{S}_l \odot [\gamma_l^D \times (\vec{\beta}^{\max} - \mathbf{D})] \\
&= \mathbf{Q}_l \mathbf{K}_l^T \odot [\mathbb{E}_n + \gamma_l^A \times \mathbf{A} + \gamma_l^D \times (\vec{\beta}^{\max} - \mathbf{D})]
\end{aligned} \tag{11}
$$

$$\mathbf{O}_l = \text{softmax}(\frac{\mathbf{S}_l'}{\sqrt{d_k}})\mathbf{V}_l \tag{12}$$

Analytically, as unveiled in Fig. 1 (b), we can dissect equation 11 through the lens of implementing multiple attention masks on the self-attention scores $\mathbf{S}_l$. Specifically, $\mathbb{E}_n \in \mathbb{R}^{n \times n}$ initially executes an unobscured mask grounded on the original attention. Concurrently, $\mathbf{A}$ could be interpreted as an additional mask that confines the current node to solely focus on its neighboring nodes. $\mathbf{D}$ could be perceived as a spatial awareness mask that adjusts the level of attention based on the distance between nodes. The significance of various masks is learned through parameters $\gamma_l^A$ and $\gamma_l^D$, which are subsequently fused to influence $\mathbf{S}_l$. By adopting this approach, we only need to introduce a minimal amount of parameters based on pure self-attention, compared to which, to achieve substantial performance enhancement. Otherwise, MSRSA shares certain similarities and associations with (Maziarka et al., 2020; Choukroun & Wolf, 2021), as they also utilize $\mathbf{A}$ and $\mathbf{D}$ to enhance the self-attention mechanism. More details on the differences and connections between them have been provided in Sec. A.3.

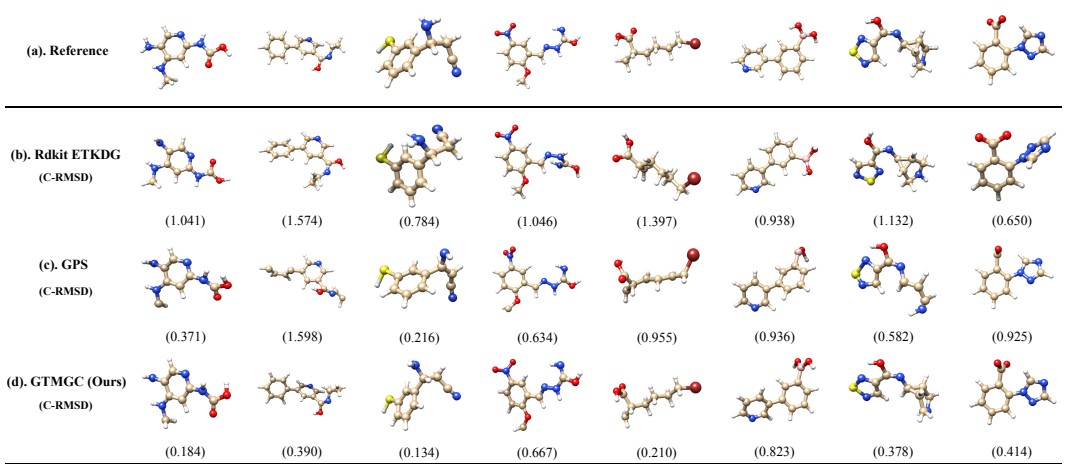

Figure 3: Examples inferred by RDkit, GPS and GTMGC (**ours**); **(a)**: The ground truth, molecules' ground-state 3D conformations; **(b)**: The predictions of RDkit; **(c)**: The predictions of GPS; **(c)**: The predictions of GTMGC. More examples are shown in Sec. C.1.

### 3.3 ARCHITECTURE OF GTMGC

**Encoder.** As shown in Fig. 2 (b), in the encoder with $N$ blocks and $H$ heads, only the adjacency matrix residual bias term $\mathbf{B}_l^A$ has been used to enhance the original self-attention mechanism. Initially, the input of the encoder only contains the node embeddings $\mathbb{H} \in \mathbb{R}^{n \times d_{model}}$ and 2D structural information, adjacency matrix, $\mathbf{A} \in \mathbb{R}^{n \times n}$ of the molecule. After forward calculation in the encoder, we pass the output of the Encoder $\mathbb{H}' \in \mathbb{R}^{n \times d_{model}}$ through a learnable weight-sharing two-layer FFN head $f_w(\cdot)$ to obtain a relatively rough conformation prediction result $\mathbf{G}_{cache} \in \mathbb{R}^{n \times 3}$, and then calculate the interatomic distance matrix of $\mathbf{G}_{cache}$, namely $\mathbf{D}_{cache} \in \mathbb{R}^{n \times n}$.

**Decoder.** Instantly, we feed $\mathbb{H}' \in \mathbb{R}^{n \times d_{model}}$, $\mathbf{A} \in \mathbb{R}^{n \times n}$ and $\mathbf{D}_{cache} \in \mathbb{R}^{n \times n}$ as inputs to the decoder with $N$ blocks and $H$ heads, as depicted in Fig. 2 (c). In the decoder, full MSRSA module with $\mathbf{B}_l^A$ and $\mathbf{B}_l^D$ bias terms has been utilized to model the molecule's spatial structure. Finally, we pass the decoder's output $\mathbb{H}''$ through the same FFN head $f_w(\cdot)$ used before to return the **final conformation** $\mathbf{G}^* \in \mathbb{R}^{n \times 3}$ predicted as result.

### 3.4 LOSS

In order to keep the prediction results rotationally and translationally invariant, we use the Mean Absolute Error (MAE) between the interatomic distance matrix of the prediction, $\mathbf{D}^*$, and that of the ground truth, $\mathbf{D}$, as the loss. In an innovative approach, we trained the model by combining the loss between its final output $\mathbf{G}^*$ and $\mathbf{G}$ with the loss between the Encoder's output $\mathbf{G}_{cache}$ and $\mathbf{G}$. This enables the Encoder to learn a more effective representation and provide the Decoder with more precise initialization space information $\mathbf{D}_{cache}$, ultimately enhancing the model's performance. The final loss function is as follows:

$$\text{MAE}(\mathbf{D}, \mathbf{D}^*) = \frac{1}{n^2} \sum_{i=1}^n \sum_{j=1}^n |\mathbf{D}_{ij} - \mathbf{D}_{ij}^*| \tag{13}$$

$$\mathcal{L} = \text{MAE}(\mathbf{D}, \mathbf{D}^*) + \text{MAE}(\mathbf{D}, \mathbf{D}_{cache}) \tag{14}$$

## 4 EXPERIMENTS

### 4.1 DATASETS

**Molecule3D.** The first benchmark introduced by (Xu et al., 2021d) that aims to use DNNs to predict the ground-state 3D geometries of molecules based solely on their 2D graph structure. The proposed large-scale dataset comprises approximately 4 million molecules, each with its own 2D molecular graph, ground-state 3D geometric structure, and four additional quantum properties. It employs two splitting methods: random splitting according to the same probability distribution and scaffold splitting based on the molecule's core component. Both partitions use a 6:2:2 ratio for training, validation, and testing.

**QM9.** A small-scale quantum chemistry dataset (Ramakrishnan et al., 2014; Wu et al., 2018) that provides geometry, energy, electronic, and thermodynamic properties for nearly 130,000 organic

molecules with 9 heavy atoms, containing molecular most stable conformation calculated by DFT. We adopt the identical data split as described in (Liao & Smidt, 2022), where 110k, 10k, and 11k molecules are allocated for training, validation, and testing, respectively.

## 4.2 METRICS

Followed (Xu et al., 2021d), given a dataset with totally $N$ interatomic distances, the Mean Absolute Error (MAE) and Root Mean Square Error (RMSE) between the prediction, $\{d_i^*\}_{i=1}^N$, and the ground truth, $\{d_i\}_{i=1}^N$, are used to evaluate the performance at node-pair level:

$$\text{D-MAE}(\{d_i\}_{i=1}^N, \{d_i^*\}_{i=1}^N) = \frac{1}{N} \sum_{i=1}^N |d_i - d_i^*| \tag{15}$$

$$\text{D-RMSE}(\{d_i\}_{i=1}^N, \{d_i^*\}_{i=1}^N) = \sqrt{\frac{1}{N} \sum_{i=1}^N (d_i - d_i^*)^2} \tag{16}$$

Additionally, We followed previous generative works (Mansimov et al., 2019; Wu et al., 2022) to calculate the Root Mean Square Deviation (RMSD) of $\mathrm{n}$ heavy atoms between the ground truth, $\mathbf{G}$, and the prediction rigidly aligned to the ground truth by the Kabsch algorithm (Kabsch, 1978), $\hat{\mathbf{G}}^*$. This metric can effectively measure the spatial difference between two conformations.

$$\text{C-RMSD}(\mathbf{G}, \hat{\mathbf{G}}^*) = \sqrt{\frac{1}{\mathrm{n}} \sum_{i=1}^{\mathrm{n}} \|\mathbf{g}_i - \hat{\mathbf{g}}_i^*\|_2^2} \tag{17}$$

## 4.3 CONFORMATION PREDICTION

**Setup.** To validate the effectiveness and advancement of our algorithm in predicting molecular ground-state conformation, we initially selected widely used **DG** and **ETKDG** algorithms implemented by RDkit as baselines. Additionally, the benchmark study (Xu et al., 2021d) employed **DeeperGCN-DAGNN** (Liu et al., 2021a) to directly predict atom coordinates, achieving previously state-of-the-art performance. Given the limited research in this specific area, we included **GINE** (Hu et al., 2019), **GATv2** (Brody et al., 2021) and **GPS** (Rampášek et al., 2022) as comparative benchmarks. GINE and GATv2, known for their impressive capabilities, are robust 2D GNNs, and GPS

Table 1: The performance on the Molecule3D and QM9 datasets (Å).

| | Validation | | | Test | | |
|---|---|---|---|---|---|---|
| | D-MAE↓ | D-RMSE↓ | C-RMSD↓ | D-MAE↓ | D-RMSE↓ | C-RMSD↓ |
| (a) Molcule3D Random Split | | | | | | |
| RDKit DG | 0.581 | 0.930 | 1.054 | 0.582 | 0.932 | 1.055 |
| RDKit ETKDG | 0.575 | 0.941 | 0.998 | 0.576 | 0.942 | 0.999 |
| DeeperGCN-DAGNN (Xu et al., 2021d) | 0.509 | 0.849 | * | 0.571 | 0.961 | * |
| GINE (Hu et al., 2019) | 0.590 | 1.014 | 1.116 | 0.592 | 1.018 | 1.116 |
| GATv2 (Brody et al., 2021) | 0.563 | 0.983 | 1.082 | 0.564 | 0.986 | 1.083 |
| GPS (Rampášek et al., 2022) | 0.528 | 0.909 | 1.036 | 0.529 | 0.911 | 1.038 |
| GTMGC (**Ours**) | **0.432** | **0.719** | **0.712** | **0.433** | **0.721** | **0.713** |
| (b) Molcule3D Scaffold Split | | | | | | |
| RDKit DG | 0.542 | 0.872 | 1.001 | 0.524 | 0.857 | 0.973 |
| RDKit ETKDG | 0.531 | 0.874 | 0.928 | 0.511 | 0.859 | 0.898 |
| DeeperGCN-DAGNN (Xu et al., 2021d) | 0.617 | 0.930 | * | 0.763 | 1.176 | * |
| GINE (Hu et al., 2019) | 0.883 | 1.517 | 1.407 | 1.400 | 2.224 | 1.960 |
| GATv2 (Brody et al., 2021) | 0.778 | 1.385 | 1.254 | 1.238 | 2.069 | 1.752 |
| GPS (Rampášek et al., 2022) | 0.538 | 0.885 | 1.031 | 0.657 | 1.091 | 1.136 |
| GTMGC (**Ours**) | **0.406** | **0.675** | **0.678** | **0.400** | **0.679** | **0.693** |
| (c) QM9 | | | | | | |
| RDKit DG | 0.358 | 0.616 | 0.722 | 0.358 | 0.615 | 0.722 |
| RDKit ETKDG | 0.355 | 0.621 | 0.691 | 0.355 | 0.621 | 0.689 |
| GINE (Hu et al., 2019) | 0.357 | 0.673 | 0.685 | 0.357 | 0.669 | 0.693 |
| GATv2 (Brody et al., 2021) | 0.339 | 0.663 | 0.661 | 0.339 | 0.659 | 0.666 |
| GPS (Rampášek et al., 2022) | 0.326 | 0.644 | 0.662 | 0.326 | 0.640 | 0.666 |
| GTMGC (**Ours**) | **0.262** | **0.468** | **0.362** | **0.264** | **0.470** | **0.367** |

*The asterisk (*) indicates that the result for this metric was not reported in (Xu et al., 2021d).

Table 2: Ablation study on input format (Å).

| | D-MAE↓ | D-RMSE↓ | ⋆ C-RMSD↓ |
|---|---|---|---|
| Ogb-style embeddings (Hu et al., 2021) | $.4299_{\pm.0014}$ | $.7162_{\pm.0003}$ | $.7431_{\pm.0198}$ |
| Atom Type IDs | $.4338_{\pm.0002}$ | $.7195_{\pm.0001}$ | $.7217_{\pm.0002}$ |
| MoleBERT Tokenized IDs (**Ours**) | $.4330_{\pm.0004}$ | $.7213_{\pm.0004}$ | $\mathbf{.7139_{\pm.0010}}$ |

⋆ means the indicator we mainly focus on.

Table 3: Ablation study of MSRSA module for ground-state conformation prediction on Molecule3D random split (Å).

| Index (ΔParam) | Methods | | | | | | D-MAE↓ | D-RMSE↓ | ⋆ C-RMSD↓ |
|---|---|---|---|---|---|---|---|---|---|
| | LPE | MHSA | $\mathbf{B}^{A}_{encoder}$ | $\mathbf{B}^{A}_{decoder}$ | $\mathbf{B}^{D(original)}_{decoder}$ | $\mathbf{B}^{D(row\text{-}sub)}_{decoder}$ | | | |
| 1 (+0) | | ✓ | | | | | $.5464_{\pm.0026}$ | $.9049_{\pm.0091}$ | $.9724_{\pm.0121}$ |
| 2 (+0) | ✓ | ✓ | | | | | $.4395_{\pm.0004}$ | $.7237_{\pm.0003}$ | $.7388_{\pm.0069}$ |
| 3 (+48) | ✓ | ✓ | ✓ | | | | $.4353_{\pm.0002}$ | $.7217_{\pm.0004}$ | $.7213_{\pm.0043}$ |
| 4 (+96) | ✓ | ✓ | ✓ | ✓ | | | $.4330_{\pm.0004}$ | $.7216_{\pm.0006}$ | $.7299_{\pm.0111}$ |
| 5 (+144) | ✓ | ✓ | ✓ | ✓ | ✓ | | $\mathbf{.4325_{\pm.0002}}$ | $.7214_{\pm.0006}$ | $.7202_{\pm.0057}$ |
| 6 (+144) | ✓ | ✓ | ✓ | ✓ | | ✓ | $.4330_{\pm.0004}$ | $\mathbf{.7213_{\pm.0004}}$ | $\mathbf{.7139_{\pm.0010}}$ |

⋆ means the indicator we mainly focus on.

is a powerful GT-style network that was recently proposed. Our model, **GTMGC**, was compared against these baselines, and the results on the Molecule3D and Qm9 datasets are presented in Table 1 (a), (b) and (c). Except for the results obtained from the original benchmark paper (Xu et al., 2021d) for DeeperGCN-DAGNN, all other findings presented in this study are derived from our experiments. It is important to note that the results of GTMGC on QM9 were fine-tuned based on its performance on Molecule3D. Finally, Fig. 3 illustrates a comparison between the predicted results of our model and the references on samples from the random split test set of Molecule3D.

**Results and Analysis.** $(i)$. As shown in Table 1 (a) and (b), in the Molecule3D random and scaffold split, GTMGC improved 18.34%, 20.86%, and 28.63% and 21.71%, 20.79%, and 22.83% respectively on the D-MAE, D-RMSE, and C-RMSD metrics in the test set compared to the previous top results. $(ii)$. Additionally, in both splits, GTMGC's results on the validation set and test set are very close, indicating that our method has good generalization performance. $(iii)$. Unlike (Xu et al., 2021d), it has already overfitted on random split and performs worse on scaffold split. GTMGC performs well on Scaffold split, indicating that it can predict molecules with different scaffolds from those unseen during training. $(iv)$. As demonstrated in Table 1 (c), GTMGC also achieved superior results on the Qm9 test set, indicating its exceptional generalization performance across various datasets and its applicability to both large and small scale molecules. $(v)$. Furthermore, as illustrated by the examples in Fig. 3, the conformations predicted by GTMGC more closely aligns with the references.

## 4.4 EFFECTIVENESS STUDY

**Setup.** To evaluate the effectiveness of GTMGC, we conducted ablation experiments on the Random Split of Molecule3D. Each result represents the mean (standard deviation, std) value, calculated after conducting the experiment 3 times. First, we modified the input format of the model while keeping the network architecture constant to evaluate the impact of our proposed input format on molecular ground-state conformation prediction. The results are presented in Table 2. Moreover, we conducted experiments to evaluate the influence of individual components within the core MSRSA module. We initially implemented a pure self-attention form and progressively incorporated various components. The outcomes of these experiments are presented in Table 3. Finally, visualization of attention weights has been employed in Fig. 4 as a means to showcase the proficiency and approach of our MSRSA module in the modeling of molecular structures. Specifically, the appendix (Sec. B Table 4) includes an extra ablation study designed to validate the contribution of each component when the MSRSA module is employed for molecular property prediction.

**Results and Analysis.** As shown in Table 2, our proposed input format, MoleBERT Tokenized IDs, outperforms the other two input formats, Ogb-style (Hu et al., 2021) embeddings and Atom Type IDs, on the C-RMSD metric which best reflects the difference between conformations and we mainly focus on. Although it falls slightly short in the D-MAE and D-RMSE indicators, the difference is not significant. However, its improvement in C-RMSD is substantial. $(i)$. This suggests that our proposed input format is more effective for ground-state conformation prediction. We

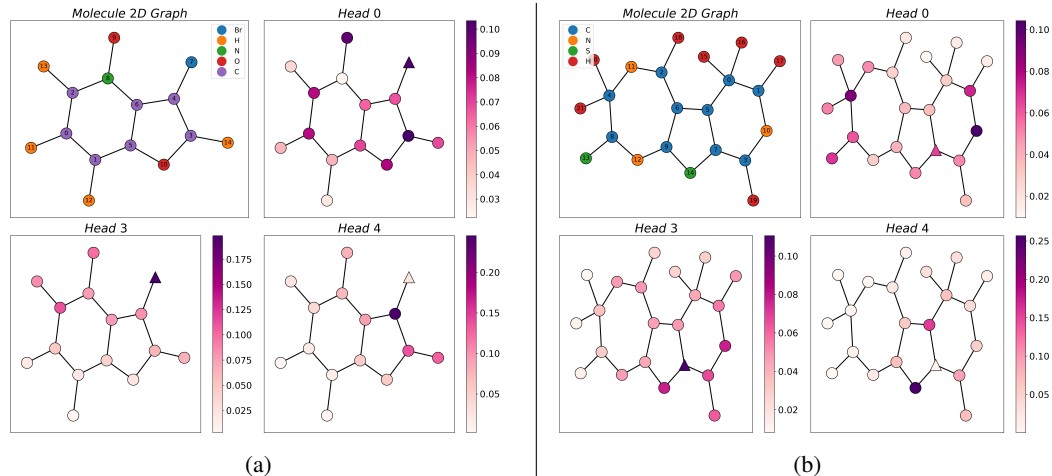

Figure 4: attention weights visualization. The triangular node ($\triangle$) represents the current node (**(a)**: No.7, Br; **(b)**: No.14, S.), while the other nodes are shaded from dark to light based on the magnitude of their attention weights relative to the current node. More examples are shown in Sec. C.2.

hypothesize that the MoleBERT Tokenizer assigns distinct IDs to atoms of the same type, resulting in embeddings with unique chemical meanings. This addresses the issue of similar type nodes having close representations after being processed by the model, leading to proximate outcomes (coordinates). Additionally, as shown in Table. 3, starting from pure self-attention, we first add $\mathbf{B}_{encoder}^{A}$ to the Encoder, followed by $\mathbf{B}_{decoder}^{A}$ to the Decoder. Finally, we incorporate $\mathbf{B}_{decoder}^{D(original)}$ and $\mathbf{B}_{decoder}^{D(row\text{-}sub)}$ into the Decoder respectively. $(ii)$. As each component is integrated, the performance of the model almost gradually improves, particularly in the C-RMSD metric, which best reflects differences between conformations and has seen improvements of 2.37%, 1.21%, and 3.37% respectively compared to pure self-attention. $(iii)$. Notably, as each component is incorporated, the standard deviation of performance (C-RMSD) gradually decreases, indicating increased stability in our model. $(iv)$. Moreover, by employing our proposed $\mathbf{D}_{row\text{-}sub}$ in place of the original inter-atomic distance matrix $\mathbf{D}$, we have achieved a performance enhancement of 0.87% along with a reduction in standard deviation. Regrettably, the $\mathbf{D}_{row\text{-}sub}$ we utilized is not sufficiently precise as it is derived from preliminary rough estimates $\mathbf{G}_{cache}$. By introducing the actual $\mathbf{D}_{row\text{-}sub}$ during molecular property prediction, superior outcomes are presented in Sec. B Table 4. $(v)$. The attention weights learned are depicted in Fig. 4, illustrating the degree of focus a specific atom has towards other atoms during its modeling. The figure reveals that the model discerns diverse attention patterns across different attention heads. For instance, in Fig. 4 (a), the atom Br, represented by a triangle, primarily captures the molecule's global information in head 0, paying substantial attention to the heavy atoms within the two rings. In heads 3 and 4, it distinctly showcases the acquisition of the molecule's spatial structural information. As observed in the figure, moving from left to right (or vice versa), and from near to far, the attention of the Br atom towards other atoms progressively diminishes. This characteristic endows our model with enhanced intuitive interpretability.

## 5 CONCLUSION

We propose a novel Transformer-based method, GTMGC, for end-to-end prediction of 3D ground-state conformations of molecules from their 2D topological structures. Our method achieves significant performance improvement over the previous best methods, reaching the state-of-the-art level, and also surpasses the widely used open-source toolkit RDkit on various metrics. Moreover, we introduce a novel and simple self-attention mechanism for molecular structure modeling, namely Molecule Structural Residual Self-Attention (MSRSA). It preserves the original self-attention structure in Transformer to a large extent, while also being able to effectively model the molecular structure. Experiments show that MSRSA module not only provides great help for predicting the ground-state conformations of molecules, but also is easy to implement and generalize to other molecular modeling tasks, such as molecular property prediction.

### ACKNOWLEDGMENTS

This work is supported by the National Natural Science Foundation of China (No.61976247) and the Fundamental Research Funds for the Central Universities (No.2682023ZTPY057).

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

# A  RELATED WORK

## A.1  MOLECULE TOKENIZATION

In natural language processing (NLP) pipelines (Vaswani et al., 2017; Devlin et al., 2018), tokenization is the first step. This involves splitting text into individual tokens. These tokens are then converted into numerical values, known as input IDs, that can be embedded by the model. With this in mind, we wondered if we could extract meaningful tokens from molecules and convert them into input IDs for our model. The most straightforward approach is to use atomic types as token IDs, but this results in a small and unbalanced dictionary size (only 118 classes of different atoms). This approach may not distinguish dominant and rare atoms in molecules, potentially causing the model to focus too much on primary atoms while ignoring rare atoms, leading to bias (Xia et al., 2023).

To our knowledge, Mole-BERT (Xia et al., 2023) is the first work to address this issue, which leverages a context-aware tokenizer based on group VQ-VAE (Van Den Oord et al., 2017) that encodes atoms into discrete values with chemical significance. This distinguishes major elements such as carbon into carbon in different chemical environments (e.g., benzene ring and aldehyde carbons.). It greatly addresses the imbalance between major and rare elements while simultaneously expanding the vocabulary size (from 118 to 512).

## A.2  MOLECULE GEOMETRY PREDICTION (GENERATION)

In recent years, there has been a surge in the application of generative models to the task of generating multiple conformations of molecules. This work primarily utilizes the GEOM-QM9 and GEOM-Drugs datasets (Axelrod & Gomez-Bombarelli, 2020). Each molecule in these datasets can exist in numerous potential stable low-energy conformations. For instance, CGCF (Xu et al., 2021a) employs a Flow model to learn the distribution $p(\mathbf{D}|\mathcal{G})$ of the interatomic distances given a molecular graph $\mathcal{G}$, and then generates potential conformations from $p(\mathbf{G}|\mathbf{D}, \mathcal{G})$ and optimizes them using Langevin Dynamics. ConvVAE (Xu et al., 2021b) adopts a bilevel programming approach to split the task into a distance prediction problem and a distance geometry problem. The entire framework encodes the molecular graph into latent space using a conditional variational autoencoder (VAE). During inference, it first samples from the latent space, then obtains sampled samples of interatomic distances through the decoder, and finally obtains the generated conformation by optimizing the distance geometry problem. ConfGF (Shi et al., 2021) suggests estimating the gradient field of the log density of interatomic distances initially, then calculating the gradient fields of the log density of atomic coordinates via the chain rule. The well-trained score network can estimate the gradient field of atomic coordinates corresponding to different levels of noise. Utilizing this capability, the annealed Langevin dynamics (Song & Ermon, 2019) algorithm can be employed to generate new conformations. However, all these methods require modeling the intermediate variable, interatomic distances, to obtain the coordinates of the generated conformation. This requirement has been identified as a contributing factor to their suboptimal performance. In an effort to circumvent the constraints posed by intermediate variables, GeoMol (Ganea et al., 2021) models essential geometric elements of molecules, such as torsion angles, bond distances, and bond angles. During inference, it outputs a set of these geometric elements, enabling the comprehensive reconstruction of 3D conformations. Moreover, numerous studies have begun to leverage the increasingly popular diffusion models (Ho et al., 2020). These models aim to learn the desired geometric distribution from a noise distribution via a reverse diffusion process. GeoDiff (Xu et al., 2022) implements this diffusion and reverse diffusion process directly on atomic coordinates. It then recovers the desired real conformation from positions sampled from the noise distribution through the reverse diffusion process. TorDiff (Jing et al., 2022), on the other hand, confines the diffusion process to the torsion angles of molecular conformations. This approach effectively reduces the dimensionality of the sampling space. A growing body of research has been advancing in accelerating the sampling speed of diffusion models, resulting in significant enhancements in both the precision and speed of molecular conformation generation (Zhang et al., 2023a; Fan et al., 2023).

**Prediction Methods.**  While generative models have demonstrated encouraging outcomes in the generation of molecular conformations, they are designed for a one-to-many task, aiming to generate multiple potential stable low-energy conformations rather than necessarily the lowest-energy ground-state conformation. In practical applications, it becomes essential to further sift through the generated results to identify conformations of superior quality and lower energy. Moreover, the inherent sampling uncertainty during the inference phase of generative models could potentially lead to the generation of conformations that are not practically feasible. Each molecule in

the GEOM (Axelrod & Gomez-Bombarelli, 2020) dataset possesses multiple low-energy stable conformations, rather than the most stable ground-state conformation. Given these challenges, a benchmark task (Xu et al., 2021d) has been proposed which seeks to predict a molecule's ground-state geometries directly from its 2D topological structure using a predictive paradigm network. A benchmark dataset called Molecule3D has been introduced which comprises nearly 4 million molecules, each with its ground-state geometric information and four quantum properties calculated at the B3LYP/6-31G* level using Density Functional Theory (DFT). Building upon this dataset, the DeeperGCN-DAGNN model is employed as a baseline by (Xu et al., 2021d) for predicting the ground-state geometry of molecules. This involves predicting the interatomic distance matrix corresponding to the molecule's ground-state conformation and directly predicting the 3D Cartesian coordinates of the molecule's ground-state conformation. Our work follows this latter approach, achieving an end-to-end prediction from a molecule's 2D graph to its 3D ground-state conformation.

### A.3 (MOLECULE) GRAPH TRANSFORMER

Transformer (Vaswani et al., 2017) networks have made remarkable strides in diverse domains (Devlin et al., 2018; Dosovitskiy et al., 2020; Liu et al., 2021d). In recent years, numerous investigations (Dwivedi & Bresson, 2020; Ying et al., 2021; Luo et al., 2022; Kim et al., 2022; Min et al., 2022) have endeavored to employ transformer networks into graphs. Studies have demonstrated that transformer networks can effectively address prevalent challenges such as over-smoothing (Chen et al., 2020) and over-squashing (Alon & Yahav, 2020) encountered in GNNs. In molecular modeling, MAT (to be discussed later) was proposed by (Maziarka et al., 2020) and R-MAT was proposed by (Maziarka et al., 2021) based on MAT. R-MAT calculates attention scores by employing distance embeddings and bond embeddings rather than using the original $\mathbf{Q}\mathbf{K}^T$. (Choukroun & Wolf, 2021) directly regulates the attention scores by incorporating a learnable gate function that considers the interatomic distances. (Wu et al., 2021a) employs two-dimensional convolution on interatomic distances for position encoding to ensure roto-translational invariance, and makes the first attempt to incorporate motifs and knowledge of functional groups into a Transformer network for 3D molecular representation learning. For a more comprehensive understanding of GTs, we recommend referring to (Min et al., 2022; Müller et al., 2023). These sources provide detailed overviews on the topic.

**Discussion.** Because $\mathbf{A}$ and $\mathbf{D}$ are the most prevalent prior knowledge for 3D molecules, numerous previous works have attempted to enhance the attention mechanism in Graph Transformer networks by leveraging these inductive biases. The works most closely related to ours are (Maziarka et al., 2020; Choukroun & Wolf, 2021). (Maziarka et al., 2020) directly incorporates $softmax(\frac{\mathbf{Q}_l \mathbf{K}_l^T}{\sqrt{d_k}})$, $\mathbf{A}$ and $g(\mathbf{D})$ throughout the model by means of three hyperparameters, where $g$ is a kernel function that can either be *softmax* or $e^{-d}$. (Choukroun & Wolf, 2021) utilizes $f(\mathbf{D}^{-1})^2$ as a gate function acting on $softmax(\frac{\mathbf{Q}_l \mathbf{K}_l^T}{\sqrt{d_k}})$, where $f$ is a learnable fully-connected network. The crucial distinction between our work and theirs lies in the fact that we do not directly incorporate $softmax(\frac{\mathbf{Q}_l \mathbf{K}_l^T}{\sqrt{d_k}})$, $\mathbf{A}$ and $\mathbf{D}$ or impose distance-based gating constraints on $softmax(\frac{\mathbf{Q}_l \mathbf{K}_l^T}{\sqrt{d_k}})$. Instead, we regard $\mathbf{S} = \mathbf{Q}_l \mathbf{K}_l^T$ as a global attention and dynamically acquire the weights of the three masks $\mathbb{E}_n$, $\mathbf{A}$, and $\vec{\beta}^{\max} - \mathbf{D}$ (corresponding to global, nearby and spatial information) for each self-attention head through learnable parameters ($1$, $\gamma_l^A$ and $\gamma_l^D$). We then adjust $\mathbf{S}$ by merging the three masks to obtain the overall mask $\mathbf{M} = \mathbb{E}_n + \gamma_l^A \times \mathbf{A} + \gamma_l^D \times (\vec{\beta}^{\max} - \mathbf{D})$, and subsequently apply softmax to the adjusted $\mathbf{S}$ to obtain the final attention weights for updating node features as $\mathbf{O}_l = \text{softmax}(\frac{\mathbf{S} \odot \mathbf{M}}{\sqrt{d_k}})\mathbf{V}_l$. Unlike (Maziarka et al., 2020), we are able to automatically learn different mask weights at every location in the model, thereby capturing distinct attention patterns. Unlike (Choukroun & Wolf, 2021), we do not directly manipulate $\mathbf{S}$ based on distance information, rather we adjust $\mathbf{S}$ using the learned mask $\mathbf{M}$. Furthermore, we introduce fewer learnable parameters. Our approach allows the model to learn different attention patterns in different self-attention heads, with some parts focusing more on global information and others on local or spatial information, thereby enhancing the expressive power of the model. A control experiment has been added to Table 4 to differentiate the variations between MAT (Maziarka et al., 2020) and our approach.

## B MSRSA FOR MOLECULAR PROPERTY PREDICTION

**Setup.** To assess the effectiveness and transferability of the MSRSA module (Sec. 3.2), We have implemented three Bert-like (encoder only) models (Devlin et al., 2018) of different sizes (small,

base and large) with complete MSRSA modules, including both $\mathbf{B}^A$ and $\mathbf{B}^D$. Detailed model hyperparameters are provided in Table 7 in Sec. D. Initially, an ablation study is conducted on our small-level model, utilizing the random split of the Molecule3D dataset. This is done to validate the effectiveness of each individual component within the MSRSA when predicting molecular properties. The results of these experiments are detailed in Table 4. Subsequently, a performance comparison was carried out with several baselines on the Molecule3D Random split for the prediction of the Homo-Lumo Gap. All baseline results are derived from (Wang et al., 2022). The results are presented in Table 5. Finally, we conduct experiments on the molecular property prediction tasks using the QM9 dataset. The results of previous methods are all borrowed from (Liao & Smidt, 2022). We use the same data split as in (Liao & Smidt, 2022), with 110k, 10k, and 11k molecules for training, validation, and testing, respectively. Our results are based on the network that was trained in

Table 4: Ablation study of MSRSA module for molecular property prediction on Molecule3D random split (eV).

| Index (ΔParam) | Methods | | | | | $\varepsilon_{\text{HOMO}}\downarrow$ | $\varepsilon_{\text{LUMO}}\downarrow$ | $\Delta\varepsilon\downarrow$ |
| | LPE | MHSA | $\mathbf{B}^A$ | $\mathbf{B}^{D(original)}$ | $\mathbf{B}^{D(row\text{-}sub)}$ | | | |
|---|---|---|---|---|---|---|---|---|
| 1 (+0) | | ✓ | | | | .209 | .239 | .281 |
| 2 (+0) | ✓ | ✓ | | | | .111 | .108 | .131 |
| 3 (+48) | ✓ | ✓ | ✓ | | | .092 | .089 | .104 |
| 4 (+96) | ✓ | ✓ | ✓ | ✓ | | .034 | **.034** | .053 |
| 5 (+96) | ✓ | ✓ | ✓ | | ✓ | **.031** | **.034** | **.045** |
| MAT (Maziarka et al., 2020) | | | | | | .067 | .070 | .083 |

Table 5: MAE results of MSRSA module for molecular HOMO-LUMO gap prediction on Molecule3D random split (eV).

| Method | Time | | $\Delta\varepsilon\downarrow$ |
| | Train | Inference | |
|---|---|---|---|
| GIN-Virtual (Hu et al., 2021) | 15min | 2min | .1036 |
| SchNet (Schütt et al., 2017) | 15min | 3min | .0428 |
| DimeNet++ (Gasteiger et al., 2020) | 133min | 16min | .0306 |
| SphereNet (Liu et al., 2021c) | 182min | 28min | .0301 |
| ComENet (Wang et al., 2022) | 22min | 3min | .0326 |
| GTMGC$_{small}$ (**Ours**) | 20min | 3min | .0446 |
| GTMGC$_{base}$ (**Ours**) | 21min | 3min | .0371 |
| GTMGC$_{large}$ (**Ours**) | 25min | 4min | .0316 |

Table 6: MAE results of MSRSA module for molecular property prediction on QM9 test set.

| Method | $\alpha$ $(a_0^3)$ | $\Delta\varepsilon$ (meV) | $\varepsilon_{\text{HOMO}}$ (meV) | $\varepsilon_{\text{LUMO}}$ (meV) | $\mu$ (D) | $C_\nu$ (cal/mol K) |
|---|---|---|---|---|---|---|
| NMP (Gilmer et al., 2017)[†] | .092 | 69 | 43 | 38 | .030 | .040 |
| SchNet (Schütt et al., 2017) | .235 | 63 | 41 | 34 | .033 | .033 |
| Cormorant (Anderson et al., 2019)[†] | .085 | 61 | 34 | 38 | .038 | .026 |
| LieConv (Finzi et al., 2020)[†] | .084 | 49 | 30 | 25 | .032 | .038 |
| DimeNet++ (Gasteiger et al., 2020) | **.044** | 33 | 25 | 20 | .030 | .023 |
| TFN (Thomas et al., 2018)[†] | .223 | 58 | 40 | 38 | .064 | .101 |
| SE(3)-Transformer (Fuchs et al., 2020)[†] | .142 | 53 | 35 | 33 | .051 | .054 |
| EGNN (Satorras et al., 2021)[†] | .071 | 48 | 29 | 25 | .029 | .031 |
| PaiNN (Schütt et al., 2021) | .045 | 46 | 28 | 20 | .012 | .024 |
| TorchMD-NET (Thölke & De Fabritiis, 2021) | .059 | 36 | 20 | 18 | **.011** | .026 |
| SphereNet (Liu et al., 2021c) | .046 | 32 | 23 | 18 | .026 | **.021** |
| SEGNN (Brandstetter et al., 2021)[†] | .060 | 42 | 24 | 21 | .023 | .031 |
| EQGAT (Le et al., 2022) | .053 | 32 | 20 | 16 | **.011** | .024 |
| Equiformer (Liao & Smidt, 2022) | .046 | **30** | **15** | **14** | **.011** | .023 |
| GTMGC$_{small}$ (**Ours**) | .131 | 47 | 37 | 32 | .055 | .050 |
| GTMGC$_{base}$ (**Ours**) | .128 | 43 | 33 | 29 | .052 | .047 |
| GTMGC$_{large}$ (**Ours**) | .117 | 39 | 29 | 26 | .043 | .043 |

† denotes using different training, validation, testing data partitions.

the previous experiment, with further simple fine-tuning performed on the Qm9 dataset. The Mean Absolute Error (MAE) between the prediction and ground truth are shown in Table 6.

**Results and Analysis.** ($i$). Table 4 robustly showcases the efficacy of each constituent within MSRSA. Notably, the gradual incorporation of components $\mathbf{B}^A$ (Index 3) and $\mathbf{B}^{D(original)}$ (Index 4), built on the foundation of pure self-attention (Index 2), has led to a stepwise enhancement in performance by 20.61% and 59.54% respectively, yet the amount of additional parameters introduced is nearly 0%. ($ii$). In a specific advancement, the substitution of the original interatomic distance matrix $\mathbf{D}$ with our innovative proposal $\mathbf{D}_{row\text{-}sub}$ (Index 4), has further boosted the effectiveness by an additional 15.09%. This underlines that $\mathbf{D}_{row\text{-}sub}$ offers superior assistance over $\mathbf{D}$ in modeling the spatial configuration of molecules. ($iii$). As illustrated in Table 5, our methodology delivers remarkable performance on large-scale molecules. Even though the large-level model encompasses a significant number of parameters, the speed of inference is still notably satisfactory. This underlines the considerable scalability of our approach when dealing with large molecules. ($iv$). The results presented in Table 6 indicate that our method attains an above-average performance in predicting three properties: $\Delta\varepsilon$, $\varepsilon_{\text{HOMO}}$ and $\varepsilon_{\text{LUMO}}$. Despite the relatively lower prediction results for the other three properties, they remain competitive when compared to numerous baseline methods. ($v$). All these results suggest that our proposed MSRSA module can be seamlessly applied to diverse molecular representation learning tasks and is characterized by its simplicity of implementation.

# C MORE EXAMPLES OF VISUALIZATIONS

## C.1 INFERENCE EXAMPLES VISUALIZATION

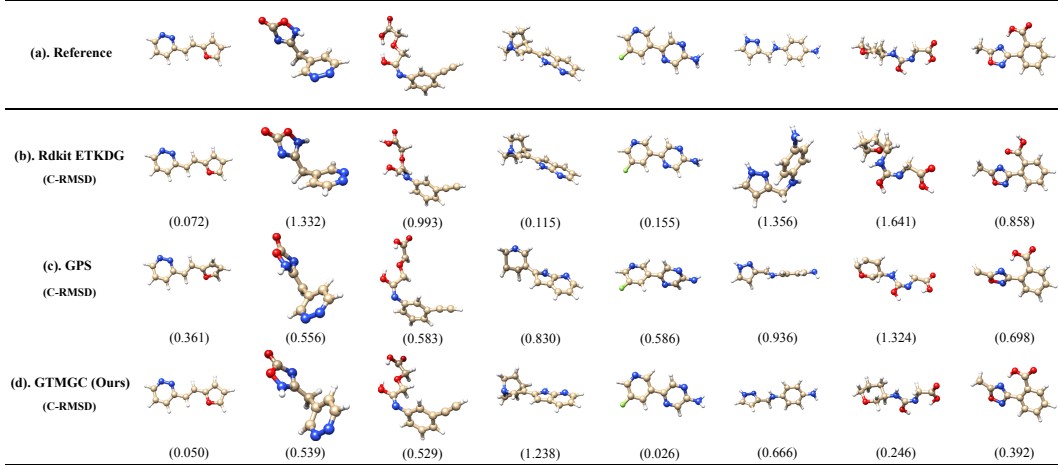

## C.2 ATTENTION WEIGHTS VISUALIZATION

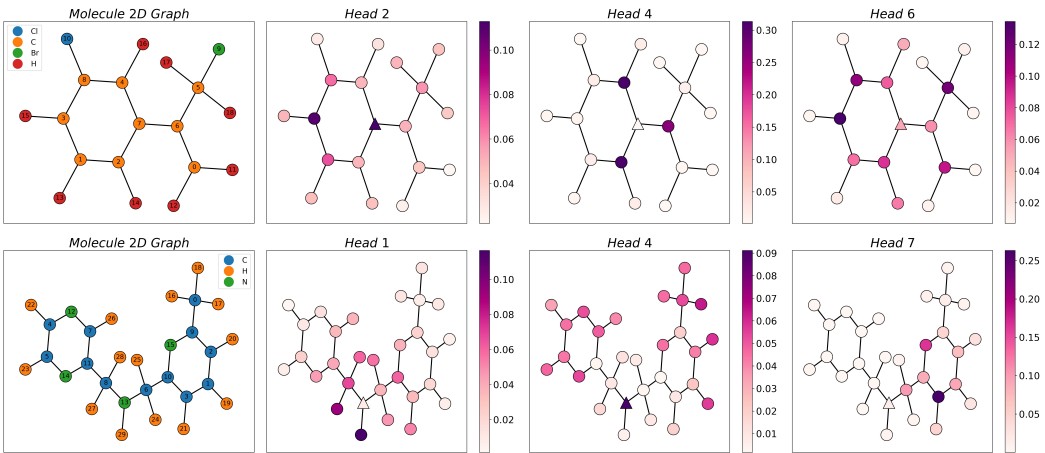

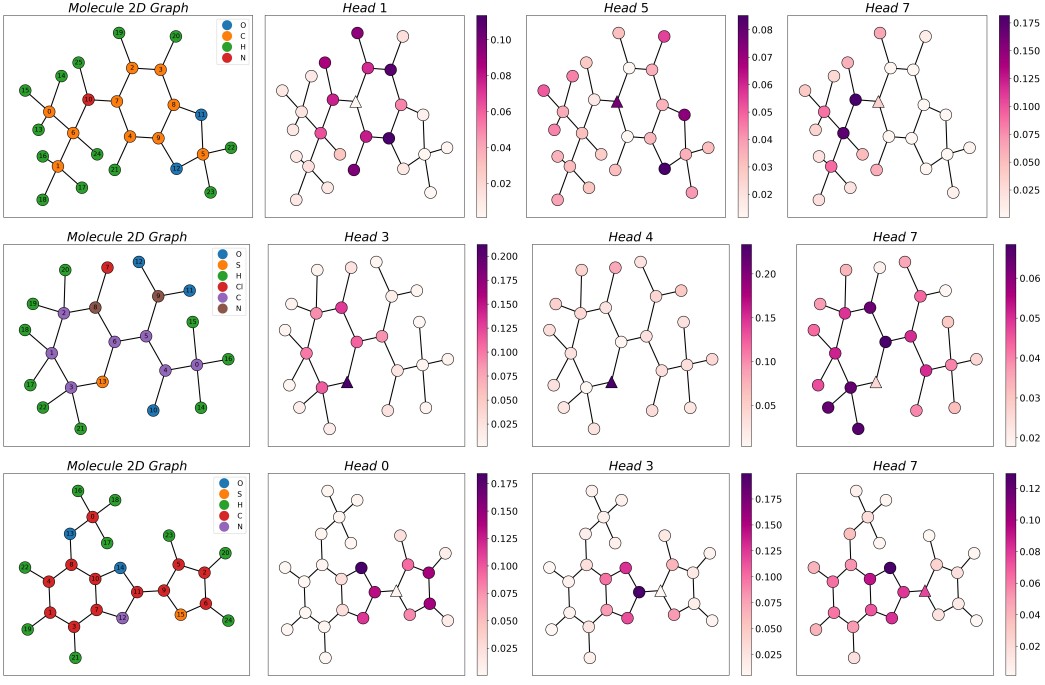

## C.3 THE LEARNED $\gamma_l^A$ AND $\gamma_l^D$

The values of $\gamma_l^A$ and $\gamma_l^D$ which we learned and exported from GTMGC$_{small}$, are illustrated in the figure below. In the experiment, only $\gamma_l^A$ and $\gamma_l^D$ are parameterized for learning, while the weight of global information is kept constant at 1. The final weights $\gamma_l^G$, $\gamma_l^A$, and $\gamma_l^D$ are determined by normalizing the values of 1, $\gamma_l^A$, and $\gamma_l^D$.

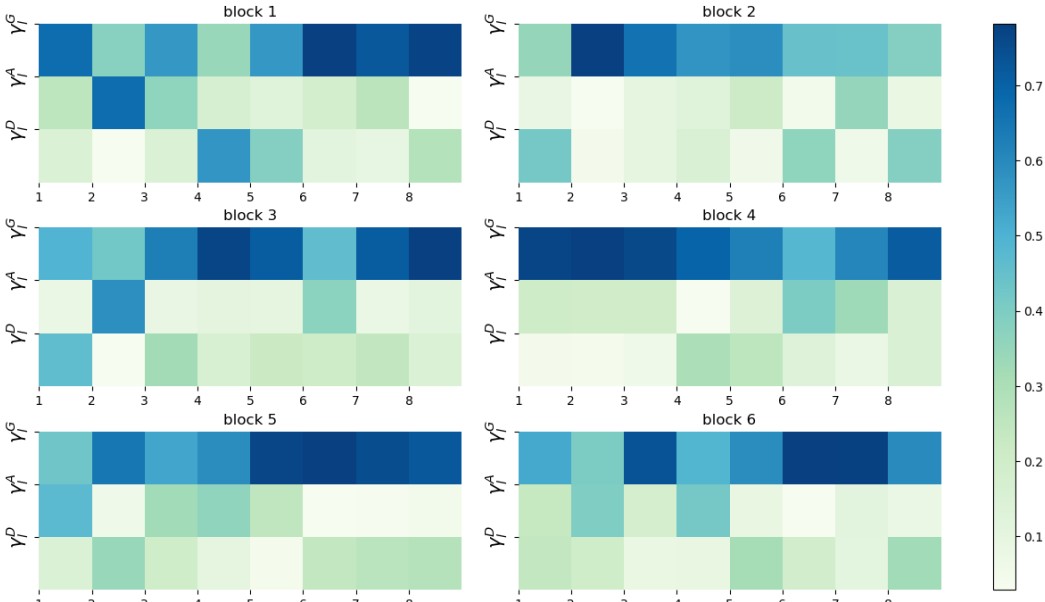

## D TRAINING DETAILS

**MoleBERT Tokenizer Training.** Since the MoleBERT tokenizer (Xia et al., 2023) is the first to tokenize atoms into a set of chemically meaningful discrete tokens, it should be re-trained on different datasets. However, once the tokenizer is trained, the tokenized results on the same dataset can be used for different tasks without changing. It is important to note that the MoleBERT tokenizer

Table 7: Model hyperparameters for GTMGC on Molecule3D

| | Ground-state Conformation Prediction | | Properties Prediction | | |
| | Encoder | Decoder | Small | Base | Large |
| --- | --- | --- | --- | --- | --- |
| Epoch | 20 | | | 60 | |
| Batchsize | 100 | | | 100 | |
| Optimizer | Adamw | | | Adamw | |
| Learning Rate | 5e-5 | | | 5e-5 | |
| Warm Up | 0.1 | | | 0.3 | |
| Lr Scheduler | linear | | | cosine | |
| Num Transformer Blocks | 6 | | 6 | 6 | 8 |
| Num Attention Heads | 8 | | 8 | 16 | 32 |
| $d\_model$ | 256 | | 256 | 512 | 512 |
| $d\_ffn$ | 1024 | | 1024 | 768 | 2048 |
| Dims in Prediction Head | [256, 768, 3] | | [256, 768, 1] | [512, 1536, 1] | [512, 1536, 1] |
| Use $\mathbf{B}^A$ | − | ✓ | | ✓ | |
| Use $\mathbf{B}^D$ | ✓ | ✓ | | ✓ | |
| Trainable Parameters | 9,808,275 (whole model) | | 4,983,393 | 11,925,185 | 26,101,889 |

is based on a small GNN model and VQ-VAE framework. We only use the tokenized tokens to train our tasks without updating the tokenizer's parameters. This is different from some GA-style (GNN layer before, Transformer layer after) approaches (Rong et al., 2020; Wu et al., 2021b; Mialon et al., 2021) that update both the GNN and Transformer layers' parameters simultaneously.

**GTMGC Training.** Table 7 presents the crucial model and training hyperparameters for GT-MGC. For our primary task of predicting the molecular ground-state conformation, both the encoder and decoder of GTMGC are configured with 6 transformer blocks and 8 attention heads. Notably, we set our $d_{model}$ to 256, $d_{ffn}$ to 1024, and $d_{hidden}$ in the prediction head to 768, resulting in a lean model with only 9M parameters. During the training phase, we employ the AdamW optimizer with a learning rate of 5e-5 and a batch size of 100. We initially warm up the learning rate from 0 to 5e-5, followed by a linear decay to 0, over a total of 20 epochs. For our auxiliary task of molecular properties prediction, we have constructed three versions of GTMGC: small, base, and large. The specific hyperparameters for these three versions are detailed in Table 7. For this task, we apply the same training strategy across all three versions of the model, but increase the number of training epochs to 60, with a larger learning rate warm up of 0.3 and a cosine decay strategy. Our open-sourced code provides further details about the training process.

