# OpenReview forum: "GTMGC: Using Graph Transformer to Predict Molecule’s Ground-State Conformation"
_ICLR.cc/2024/Conference — ICLR 2024 spotlight_

### Official Review · Reviewer_1WnG · 2023-10-25

**Soundness:** 3 good
**Presentation:** 3 good
**Contribution:** 2 fair
**Rating:** 3
**Confidence:** 4

**Summary:**

The paper proposes GTMGC, a transformer-based architecture for end-to-end prediction of 3D groundstate molecules' conformations from their 2D graph.

The method makes use of MoleBERT for initial embedding as well as LPE for positional encoding.
The inputs are then processed by a Transformer-based model, where the self-attention modules are augmented with the adjacency matrix and learned/predicted atomic distance matrix in a weighted sum fashion.

The loss is augmented by a regularization of the middle distance matrix prediction.,

**Strengths:**

1. The paper is very clear and pleasant to read.
2. The integration of various existing frameworks for the task of molecules' ground state conformation is original to some extent.
3. The performances compared to the baselines are significant.

**Weaknesses:**

I believe there are two main weaknesses:

1. Novelty: There are plenty of previous works (not cited) implementing Transformer architectures that implement the elements of the proposed self-attention, e.g., [1] uses the adjacency matrix, [2] makes use of the distance matrix (even cited in the manuscript), while the weighted summation is ubiquitous in many fields using Transformers [3]. The initial encoding is obviously not a technical contribution.

Thus, the contribution may be summarized as the integration of existing methods/approaches, augmented with the regularization loss on the distance matrix (the \beta seems to bring very minor improvement).

2. Strange results: I may be wrong, but according to Table 3 (ablation study) a *simple Transformer* architecture without any addition to the self-attention reaches 0.4395 (MAE) which is already better than all the other baselines. This is problematic. Also, the final improvement is only ~1%. Finally, the reported results seem to take the best-ablated model results for each metric, which is wrong.

Moreover,  we have:

3. The advantage of MoleBert over the standard atomic encoding is extremely shallow or even worse.
4. Lack of comparison with other Transformer based methods.

[1] *Molecule attention transformer.*

[2] *Geometric transformer for end-to-end molecule properties prediction.*

[3] *Axial-DeepLab*

**Questions:**

Currently, it seems the proposed contributions don't bring any advantage over a simple (large) transformer model.

1. One needs to know the capacity of the model in order to assess the origin of the good performance.
According to weakness 2, it seems the good performances are obtained *almost solely* from a large/powerful standard Transformer model. From Table 5, the model seems much bigger than other methods. Also, the discrepancies between the tables are disturbing (one single metric should be used for the model validation).

2. Ablating the initial LPE.

3. It would be beneficial to have a comparison performance with (at least one) other molecule transformer-based methods such as [4,5,6] or others at a similar capacity.

4. It would be interesting to look at the learned weighting parameters (\gamma) of the self-attention to better understand the contribution of each (maybe even adding a weighting to the global attention).


[4] Relative molecule self-attention transformer.

[5] 3dtransformer: Molecular representation with transformer in 3d space.

[5] Geometric transformer for end-to-end molecule properties prediction.

---

> ### Author Response · Authors · 2023-11-17
> **Response to Reviewer 1WnG's Weakness1**
>
> Dear Reviewer,
>
> We greatly appreciate you taking the time to review our manuscript. Your feedback and suggestions are extremely valuable to us and will help us improve and enhance our work.
>
> **Weakness 1:**
>
> > Novelty: There are plenty of previous works (not cited) implementing Transformer architectures that implement the elements of the ......
>
> **Response:**
>
> As you mentioned, numerous studies have attempted to utilize $\mathbf{A}$ and $\mathbf{D}$ to enhance the Transformer network structure in molecular modeling. We believe that $\mathbf{A}$ and $\mathbf{D}$  are the most prevalent inductive biases for 3D molecular structures. Therefore, our innovation lies not in augmenting self-attention with A and D per se, but in the manner we integrate A and D into the network.
>
> First, we list our core formulas and the core formulas of the two works [1] [2]:
>
> **Ours**:  $\mathbf{S}_l^\prime=\mathbf{Q}_l{\mathbf{K}_l^T}\odot{[\mathbb{E}_n+{\gamma_l^A\times{\mathbf{A}}}+\gamma_l^D\times{(\vec{\beta}^{\max}-\mathbf{D})}]}$ and $\mathbf{O}_l=\text{softmax}(\frac{\mathbf{S}_l^\prime}{\sqrt{d_k}})\mathbf{V}_l$, where $\gamma_l$ is learnable parameters.
>
> **[1]**: $\mathbf{O}_l=(\lambda_a\text{softmax}(\frac{\mathbf{Q}_l{\mathbf{K}_l^T}}{\sqrt{d_k}})+\lambda_dg(\mathbf{D})+\lambda_g\mathbf{A})\mathbf{V}_l$, where $\lambda$ is hyperparameters and $g$ is a function.
>
> **[2]**: $\mathbf{O}_l=\text{softmax}(\frac{\mathbf{Q}_l{\mathbf{K}_l^T}}{\sqrt{d_k}})\odot{f({\mathbf{D}}^{-1})^2}\mathbf{V}_l$,  where $f$ is a shallow, fully connected neural network.
>
> Although our method has certain similarities with theirs, we emphasize the following differences to enhance our novelty：
>
> 1. Figure 1(b) in our manuscript demonstrates our main idea. We propose using three masks, namely $\mathbb{E}_n$, $\mathbf{A}$, and $\vec{\beta}^{\max}-\mathbf{D}$, and adjusting the different combinations between each head with learnable parameters $\gamma_l^A$ and $\gamma_l^D$. Ultimately, we aim to adjust $\mathbf{Q}_l{\mathbf{K}_l^T}$ according to the final attention-mask, ${\mathbb{E}_n+{\gamma_l^A\times{\mathbf{A}}}+\gamma_l^D\times{(\vec{\beta}^{\max}-\mathbf{D})}}$ , that has learned and integrated ‘global’, ‘nearby’, and ‘spatial’ information **in each attention head **. This approach differs significantly from [1]. In [1], the self-attention mechanism is also enhanced through the use of $\mathbf{A}$ and $\mathbf{D}$ as biases. However, [1] directly adds $\mathbf{A}$ and $g(\mathbf{D})$ to the attention-weights $softmax(\frac{\mathbf{Q}_l{\mathbf{K}_l^T}}{\sqrt{d_k}})$ with global hyperparameters $\lambda$ which means they can't learn diffrent attention patterns (as dipected in Figure 4 in our manuscript) across different  attention heads.
> 2. We have innovatively adopted the $\vec{\beta}^{\max}-\mathbf{D}$ approach to incorporate the hypothesis that "the farther the distance, the smaller the interaction". This approach has shown some improvement in performance in both table 3 and table 4 of our manuscript when compared to the original $\mathbf{D}$ approach, although not in all cases. It is worth noting that this approach bears some similarity to $g(\mathbf{D})$ in [1] and $f({\mathbf{D}}^{-1})^2$ in [2] with $f$ being a neural network.  But in terms of details, the importance we place on “distance” varies in different places, but [1] is the same everywhere. In addition, [2] may have parameterized different $f$ in each Trans block, but in each block, the importance they focus on  “distance” for different attention heads is still the same. And we only introduce h (num attention heads) learnable parameters in each block, while [2] introduces a fully connected neural network.
> 3. We strictly follow that the attention-weights $\mathbf{O}_l=\text{softmax}(\frac{\mathbf{S}_l^\prime}{\sqrt{d_k}})$ are normalized, but they might not have.
>
> We wish that the aforementioned explanation angle could enhance our novelty. To summarize, our innovative approach to introduce $\mathbf{A}$ and $\mathbf{D}$ into self-attention is that:
>
>  In each attention head, we capture different attention patterns by automatically learning the weights among “global”, “nearby”, and “spatial” information. The final attention-mask $\mathbb{E}_n+{\gamma_l^A\times{\mathbf{A}}}+\gamma_l^D\times{(\vec{\beta}^{\max}-\mathbf{D})}$, which is obtained after fusion, is used to adjust the attention-score $\mathbf{Q}_l{\mathbf{K}_l^T}$. Then, through softmax, we obtain the final attention-weights $\text{softmax}(\frac{\mathbf{S}_l^\prime}{\sqrt{d_k}})$  to weight the node features $\mathbf{V}_l$. The key point is that the patterns learned in each attention head are different. We don’t need to adjust additional hyperparameters and introduce very few learnable parameters to get performance improvement (as demonstrated in tables 3 and 4). Moreover, it allows for a more interpretable attention map (as depicted in Figure 4).

---

> > ### Author Response · Authors · 2023-11-17
> > **Response to Reviewer 1WnG's Weakness1**
> >
> > At the same time, we have also supplemented additional chapters in the related works (Sec. A.1.3) to complement the similarities and differences between us and these works, in order to make a better distinction and highlight our innovation.
> >
> > We have also added additional comparative experiments in Table 4 (MAT) to demonstrate the superiority of our method over [1] and because [2] did not use A, so we did not have a comparison. (To ensure a fair comparison between our method and the self-attention method in [1], we directly replaced the attention formula in [1] in our code to keep the other settings the same.)
> >
> > [1] Molecule attention transformer.
> >
> > [2] Geometric transformer for end-to-end molecule properties prediction.

---

> ### Author Response · Authors · 2023-11-17
> **Response to Reviewer 1WnG's Weakness2&3&4**
>
> **Weakness 2:**
>
> > Strange results: I may be wrong, but according to Table 3 (ablation study) a *simple Transformer* architecture without any addition to the self-attention reaches 0.4395 (MAE) which is already better than all the other baselines. This is problematic. Also, the final improvement is only ~1%. Finally, the reported results seem to take the best-ablated model results for each metric, which is wrong.
>
> **Response:**
>
> There is indeed such a problem, and the simple Transformer structure has indeed achieved good results. However, from an application standpoint, it is also considered an "innovation" to use such a structure for the first time in completing this task. What we would like to emphasize more is our focus on the C-RMSD indicator (as mentioned in our manuscript) since it is widely used in the field of bioinformatics to measure the difference between two spatial structures, thus making it more convincing. The D-MAE and D-RMSE indicators are derived from the Molecule3D benchmark. In order to ensure a fair comparison, we still use these two indicators as references. Overall, our proposed structure has demonstrated an improvement of more than 3% in C-RMSD compared to Simple Trans, thereby proving its effectiveness. It is possible that this task is more difficult to optimize, hence we have strongly demonstrated in Table 4 that our proposed module has been successfully applied to molecular property prediction, resulting in significant performance improvements compared to simple Trans.
>
> In addition, we corrected our careless error reporting results in Table 1 (best model on Molecule3D Random Test, D-MAE: 0.4325; D-RMSE:0.7210; C-RMSD:0.7129).
>
> **Weakness 3 & 4:**
>
> > The advantage of MoleBert over the standard atomic encoding is extremely shallow or even worse.
> >
> > Lack of comparison with other Transformer based methods.
>
> **Response:**
>
> As with the response to Weakness2, we are more focused on the results on C-RMSD. The introduction of MoleBert indeed brought improvements on C-RMSD, so we retained it as a small innovation.  In addition, we have taken into account the opinions of another reviewer and added a set of comparison ablation experiments in Table 2 for another input format (Orb-style embeddings). The results show that its performance on C-RMSD is far inferior to our method.
>
> We compared our method with the GPS method in Table 1, which is a Graph Transformer (GT) method  (as mentioned in Sec. 4.3). In Table 6, both SE(3)-Transformer and Equiformer are GT methods. And also MAT in table4.

---

> ### Author Response · Authors · 2023-11-17
> **Response to Reviewer 1WnG's Questions**
>
> **Question 1:**
>
> > One needs to know the capacity of the model in order to assess the origin of the good performance. According to weakness 2, it seems the good performances are obtained *almost solely* from a large/powerful standard Transformer model. From Table 5, the model seems much bigger than other methods. Also, the discrepancies between the tables are disturbing (one single metric should be used for the model validation)
>
> **Response:**
>
> As for the responses to Weakness 2&3, Pure Trans indeed has brought very good performance. However, as shown in the updated Table 3 and Table 4, our MSRSA module introduces extremely few extra parameters (close to 0%) and achieves 3% and 60% performance improvements on the main task, ground-state conformation prediction, and the transfer task, molecular property prediction, respectively.
>
> Here, we express our apologies. We used wrong calculation results in Table 5 of our previous manuscript. Therefore, we reorganized the performance of the MSRSA module on molecular property prediction in Table 5 and Table 6 in Sec. A.2. The new correct results show that we achieved considerable results on Molecule3D with fast inference time. The results on Qm9 also achieved a medium-to-high level on Gap, Homo, and Lumo.
>
> While our $GTMGC_{large}$ does have a relatively large parameter size, it introduces less inductive bias compared to other baselines (bond angles, dihedral angles and so on). Moreover, its inference speed is impressively efficient.
>
> **Question 2:**
>
> > Ablating the initial LPE.
>
> **Response:**
>
> we add the ablation experiments on LPE  in Table 3 and Table 4. However, the results show that the performance is very poor after removing LPE. This is because PE is an important component in the Transformer network (as is the case in NLP). Therefore, we regard PE as a member of Pure Trans.
>
> **Question 3:**
>
> > It would be beneficial to have a comparison performance with (at least one) other molecule transformer-based methods such as [4,5,6] or others at a similar capacity.
> >
> > [4] Relative molecule self-attention transformer.
> >
> > [5] 3dtransformer: Molecular representation with transformer in 3d space.
> >
> > [6] Geometric transformer for end-to-end molecule properties prediction.
>
> **Response:**
>
> Due to time and resource constraints, we have added a comparison experiment with [2] in Table 4 (MAT). We did not fully reproduce the model in [2], but instead directly replaced the attention formula in our code with MAT to corroborate the response to weakness1.
>
> **Question 4:**
>
> > It would be interesting to look at the learned weighting parameters (\gamma) of the self-attention to better understand the contribution of each (maybe even adding a weighting to the global attention).
>
> **Response:**
>
> The learned weights $\gamma$ are demonstrated in Section A.3.3. The corresponding weights are obtained and displayed by normalizing (1, $\gamma_l^A$ and $\gamma_l^D$).
>
> Please let us know if you have any further questions or concerns. If we have addressed your concerns, we would appreciate it if you could consider increasing the score.

---

> ### Author Response · Authors · 2023-11-21
> **Urgent Request for Reviewer 1WnG's Response**
>
> Dear Reviewer 1WnG,
>
> We apologize for the interruption! We understand that you may be very busy, but we hope that you can take some time to review and respond to our rebuttal. As the rebuttal phase is about to end, we believe that our response can adequately address your doubts and questions to a certain extent, and we have made corresponding modifications based on your suggestions.
>
> We very much hope to receive your reply, which is very important for the improvement of our paper. If you have any other questions, we hope you can communicate with us again.
>
> Looking forward to your reply.
>
> Thank you!
>
> Best regards,
> Authors of Paper ID 5387

---

> > ### Comment · Reviewer_1WnG · 2023-11-21
> >
> > Thank you for your rebuttal and detailed answers.
> >
> > Weakness 1: I am aware of the differences in formulation. I meant that the proposed method integrates existing self-attention approaches, which greatly reduces the paper's novelty.
> >
> > Baselines: The problem with MAT is that it is the worst transformer-based model by very large margins. I suggested a few but there are many other Transformer based works with much better performance. Table 3 shows a very low 3% improvement.
> > Also, Table 6 shows very bad results (in fact almost every existing Transformer-based prediction model would beat it) in molecular property prediction compared to the baselines while having a much higher model capacity.
> > Given Table 6, the new jump in performance in Table 4 is very strange and hard to understand.
> >
> > Question 4: This is a new very interesting analysis. We can observe $\gamma^D$ is almost always the lowest and close to zero (not sure why they have been normalized). It does not really support the importance of its contribution. How do you explain that?

---

> > > ### Author Response · Authors · 2023-11-21
> > > **Follow-up Response**
> > >
> > > Dear Reviewer,
> > >
> > > We are very grateful for your reply.
> > >
> > > **Response to Weakness 1:**
> > >
> > >  We do not believe that our approach is an integration of existing methods. Our perspective is different, our innovation is to adjust the self-attention score by automatically learning the importance between “global”, “nearby”, and “spatial” information. Although the resulting formula form is similar, its starting point is different. And the experiment proved its effectiveness.
> > >
> > > We did a comparison with MAT to show that our simple improvement on its basis achieved a great improvement.
> > >
> > > Although our results in Table 6 are not very good, we are not aiming to achieve SOTA, we just want to show that our proposed simple MSRSA module can be quickly applied to other molecular modeling tasks. In addition, Table 5 also clearly shows that our method has good performance on large molecules, although our parameter size is relatively large, but its speed is not slow. I think our idea is to introduce as little inductive bias as possible to achieve good results, which leads to the inevitability of large model capacity.
> > >
> > > Indeed, many existing GT methods may outperform us, but they often involve complex and cumbersome designs. In contrast, we place emphasis on the simplicity and user-friendliness of our approach, which allows us to achieve respectable results. This balance between ease of use and performance is a key aspect of our methodology.
> > >
> > > Table 4 is the ablation experiment we did again with the small scale model, which is strictly trained to convergence. Table 6 is a simple fine-tuning, without rigorous parameter tuning training, just to show that it can achieve good performance when applied to other tasks. We are also surprised that its performance on large data is better than fine-tuning on small data sets.
> > >
> > > **Response to Question 4:**
> > >
> > > This is a good question. But we think this does not mean that it is not enough to support the importance of its contribution. In our view, self-attention already has a strong ability to model the relationship between nodes to a certain extent. Therefore, $D$ should be regarded as a bias term that introduces spatial information, and the degree to which the model needs to rely on it to achieve better performance only needs to be controlled by the learned $\gamma_l^D$. As shown in Table 4, the effect is significantly improved after introducing $D$ through $\gamma_l^D$. Although in the visualization, most of the $\gamma_l^D$ are relatively small, but it does not mean that introducing $D$ through $\gamma_l^D$ is meaningless.
> > >
> > > Thank you very much for your reply despite your busy schedule, we look forward to more of your feedback!
> > >
> > > Best regards, Authors

---

### Official Review · Reviewer_Po8N · 2023-10-30

**Soundness:** 3 good
**Presentation:** 3 good
**Contribution:** 3 good
**Rating:** 8
**Confidence:** 4

**Summary:**

This work proposes GTMGC, a graph transformer model for ground-state molecular conformation prediction. GTMGC uses a novel self-attention module to achieve effective molecular structure modeling. Experiments show that the proposed GTMGC model achieves state-of-the-art performance in ground-state molecular conformation prediction benchmarks.

**Strengths:**

Originality: The proposed graph transformer model is novel, with many novel technical contributions in effectively capturing spatial structures by self-attention mechanism.
Quality: The effectiveness has been effectively demonstrated by experiments.
Clarify: The writing and presentation of this paper is good and well-organized.
Significance: The contribution of this work is very useful and meaningful to chemical and molecular biological science fields as the proposed method can significantly accelerate the computation of finding ground-state molecular conformations.

**Weaknesses:**

There are actually many prior studies about formulating the mapping from 2D molecular graphs to 3D molecular conformations as a generative problem. Though they are different from the problem studied in this work, these models can be trained on the used Molecule3D datasets and evaluated by generating only one molecular conformation. However, authors do not compare with any of these methods. Authors are recommended to compare with at least one molecular conformation generation method, such as [1].

[1] Torsional Diffusion for Molecular Conformer Generation. NeurIPS 2022.

**Questions:**

No additional questions.

---

> ### Author Response · Authors · 2023-11-17
> **Response to Reviewer Po8N**
>
> Dear Reviewer,
>
> We would like to express our sincere gratitude for the time and effort you have dedicated to reviewing our manuscript.
>
> **Weakness 1:**
>
> > There are actually many prior studies about formulating the mapping from 2D molecular graphs to 3D molecular conformations as a generative problem. Though they are different from the problem studied in this work, these models can be trained on the used Molecule3D datasets and evaluated by generating only one molecular conformation. However, authors do not compare with any of these methods. Authors are recommended to compare with at least one molecular conformation generation method, such as [1].
> >
> > [1] Torsional Diffusion for Molecular Conformer Generation. NeurIPS 2022.
>
> **Response:**
>
> Indeed, a substantial amount of contemporary research focuses on using generative models to generate multiple potential conformations with low energy, conditioned on the 2D graph structure of molecules. As you correctly pointed out, Molecule3D could potentially utilize these models for a comparison in single-generation.
>
> However, it is unfortunate that the currently effective algorithms for conformation generation, including TorDiff[1], ConfGF[2], GeoDiff[3], among others, mainly rely on score-based models and diffusion models. These models require multiple iterations during the sampling process. For example, GeoDiff necessitates a denoising process of 5000 steps to sample a single instance. This approach is impractical for our dataset, which consists of 700,000 large-scale molecules in the test set. To the best of our knowledge, these generative benchmarks only cover a mere 200 molecules within the test set.
>
> [1] Torsional Diffusion for Molecular Conformer Generation
>
> [2] Learning Gradient Fields for Molecular Conformation Generation
>
> [3] GeoDiff: a Geometric Diffusion Model for Molecular Conformation Generation
>
> Please let us know if you have any further questions or concerns.

---

> > ### Comment · Reviewer_Po8N · 2023-11-18
> > **Follow-up Response**
> >
> > I understand that some conformation generation methods, such as ConfGF and GeoDiff, requires thousands of diffusion steps to generate molecular conformations. Nonetheless, some other methods may not require such a large cost. According to Section 4.3 of Torsional Diffusion paper [1], Torsional Diffusion requires only 5~20 diffusion steps for generation, and the runtime of Torsional Diffusion and GeoMol [2] is not high compared with RDKit (Table 2). Do authors think it is computationally practical to run Torsional Diffusion or GeoMol on your datasets?
> >
> > [1] Torsional Diffusion for Molecular Conformer Generation. NeurIPS 2022.
> > [2] GeoMol: Torsional Geometric Generation of Molecular 3D Conformer Ensembles. NeurIPS 2021.

---

### Official Review · Reviewer_BbTi · 2023-10-31

**Soundness:** 3 good
**Presentation:** 3 good
**Contribution:** 3 good
**Rating:** 8
**Confidence:** 4

**Summary:**

This paper introduces a novel graph transformer specifically designed for 3D ground state prediction. Notably, the proposed architecture is versatile and applicable to a wide range of 3D supervised tasks. The key contributions of this work include:

A novel architectural proposal that elegantly extends the classical attention mechanism to 3D molecular graphs. This extension incorporates edge and interatomic distances as biases for the attention mechanism, enhancing its capabilities.

A successful demonstration of the effectiveness of this architecture in the realm of 3D ground state prediction, as well as its application to predict various other 3D molecular properties.

**Strengths:**

Originality:
Despite numerous unsuccessful attempts to construct graph transformers using principles akin to the original transformer, this work stands out as it elegantly achieves the intended goal with minimal architectural complexity. This work avoids unnecessary biases that often detract from the model's effectiveness, making it easier for most researchers to apply their existing intuitions from sequence transformers to this novel architecture.

Quality:
The architectural design and its application in ground state conformation prediction are well executed, as reflected in the results, solidifying its position as a favorable solution compared to alternative methods. The iterative refinement of $G_{cache}$ within the decoder represents a notable innovation that enhances model performance in conformer prediction. Ablation studies further clarify the significance of each component within the network, facilitating an understanding of their contributions to this specific modeling task.

Clarity:
The paper is well-written and maintains a high level of clarity, making it easily comprehensible for readers.

Significance:
While predicting the ground state of a molecule remains relatively underexplored due to its limited relevance in specific applications, this work serves as a foundational step that can be extended to tackle the broader challenge of full conformer generation. Such an extension holds are very significant, especially in the context of drug discovery.

**Weaknesses:**

The assertion regarding the innovative utilization of the MoleBERT Tokenizer might be overstated, especially in light of the results presented in Table 2. Previous molecular graph papers, such as the MolGPS paper, have explored various atomic featurizations that could potentially outperform the approach presented in this work.

To allocate more space for related works and experiments, it would be beneficial to consider shortening or omitting certain sections, such as those in the introduction (implementation) and preliminary sections (multi-head and transformer).

The related work should be integrated into the main text rather than relegated to the appendix. It is crucial to comprehensively cover the various attempts to construct graph transformers and elucidate why they are ill-suited for the tasks at hand.

**Questions:**

Was molecular property prediction approached as a single-task or multi-task endeavor?

Could you clarify the rationale behind placing the molecular property prediction results in the appendix, especially considering that they do not outperform SOTA across the board?

It could be valuable to assess the scalability of your architecture across various graph sizes, thereby determining where it potentially outperforms existing methods.

---

> ### Author Response · Authors · 2023-11-17
> **Response to Reviewer BbTi**
>
> Dear Reviewer,
>
> We would like to express our sincere gratitude for the time and effort you have dedicated to reviewing our manuscript. Your comments and suggestions have been invaluable in improving the quality of our work. In the following sections, we will address each of the points you have raised in your review. We hope that our responses and the revisions we have made will meet your approval.
>
> **Weakness 1:**
>
> > The assertion regarding the innovative utilization of the MoleBERT Tokenizer might be overstated, especially in light of the results presented in Table 2. Previous molecular graph papers, such as the MolGPS paper, have explored various atomic featurizations that could potentially outperform the approach presented in this work.
>
> **Response:**
>
> Thank you for your valuable comments! We would like to clarify that our primary focus in this task is the C-RMSD evaluation metric, which is a widely accepted measure in bioinformatics that quantifies the 3D spatial difference between two structures. The D-MAE and D-RMSE metrics, which were derived from the Molecule3D benchmark, were incorporated to facilitate a fair comparison. The introduction of MoleBert, particularly in comparison with atom ids, has significantly enhanced our performance on the C-RMSD metric. We consider this to be a noteworthy innovation in terms of performance improvement.
>
> In response to your suggestion, we’ve conducted an additional  ablation experiment using Ogb-style embeddings. These embeddings are node features parsed by rdkit and are represented as a vector of length 9, with each element signifying a different atomic property. The outcomes can be found in the updated version of Table 2. While this method of atomic featurization shows good results in terms of D-MAE and D-RMSE, it unfortunately falls short when evaluated against the C-RMSD indicator. Despite this setback, we maintain our belief that the Molebert Tokenized IDs, which we proposed, offer a beneficial approach to predicting molecular conformation.
>
> **Weakness 2:**
>
> > To allocate more space for related works and experiments, it would be beneficial to consider shortening or omitting certain sections, such as those in the introduction (implementation) and preliminary sections (multi-head and transformer).
> >
> > The related work should be integrated into the main text rather than relegated to the appendix. It is crucial to comprehensively cover the various attempts to construct graph transformers and elucidate why they are ill-suited for the tasks at hand.
>
> **Response:**
>
> According to your suggestions, we have appropriately discussed some relevant literature in the Introduction section. However, considering the natural flow of the text and the impact of the notation and formulas in the preliminary sections on the subsequent Method chapter, as well as the limited amount of work directly related to our task, ground-state conformation prediction (which is not widely studied), we have still chosen to present it in the Appendix.
>
> Nonetheless, we have also included an additional section on the related work of "graph transformers". Regarding the feasibility of other GT methods, we mainly emphasize the simplicity and elegance of our structure (as mentioned in the text), which has already demonstrated good performance. We have also compared it with GPS, a kind of GT method, in Table 1.
>
> **Question 1:**
>
> > Was molecular property prediction approached as a single-task or multi-task endeavor?
>
> **Response:**
>
> Single-task.
>
> **Question 2:**
>
> > Could you clarify the rationale behind placing the molecular property prediction results in the appendix, especially considering that they do not outperform SOTA across the board?
>
> **Response:**
>
> The purpose is to illustrate that our proposed MSRSA module can be easily migrated to molecular property prediction and achieve remarkable results. Although not outstanding, our structure is simpler and easier to implement.
>
> We would like to apologize and clarify that the data used in Table 5 of our previous manuscript was incorrectly calculated. As a result, we have made corrections in Sec. A.2 and reorganized our molecular property prediction experiments in Tables 4, 5, and 6. The experimental results show:
>
> 1. The ablation study validates the effectiveness of the components of the MSRSA module.
> 2. Table 5 demonstrates that our method achieves considerable performance on large-scale macromolecular datasets while maintaining desirable inference speed.
> 3. On Qm9, we achieved above-average performance on three properties: Gap, Homo, and Lumo. The performance on the other three properties is not significantly different from other baselines.
>
> This fully demonstrates that our structure, while simple and elegant, can be quickly transferred to other tasks and achieve good performance.
>
> Please let us know if you have any further questions or concerns. Thanks again!

---

### Meta-Review · Area_Chair_QJV7 · 2023-12-05

**Metareview:**

The paper addresses the problem of directly predicting the ground-state 3D configuration of a molecule from its 2D topological description.
This problem is currently solved using computationally expensive quantum chemistry techniques such as DFT, and speeding up these calculations does have the potential to significantly benefit areas such as drug discovery and materials science.

The evaluation, in order to compare against a wide variety of existing work sets itself the goal of "first predict 3D, then predict molecular properties".
This is very reasonable, and indeed one might not expect SOTA compared to methods which directly aim to predict the properties.

Strengths:
 - Reviewers agree that the model is novel, that it addresses a fundamental problem, and that it does so in an elegant manner
 - Directly addressing the problem of predicting conformers has the potential to be applied to a variety of downstream tasks.
 - Model achieves SOTA performance on its core task.

Non-weaknesses:
 - I agree with the authors that it is not necessary to reach or exceed SOTA on molecular property prediction tasks, which are essentially "cross-checks" that the core task is useful for tasks on which previous ML approaches are known to work.
 - I agree with the authors that the additional parameters are not in themselves a problem, that the key is (a) whether the model generalizes as expected, (b) that the computational burden is "reasonable", where that can be defined as "considerably cheaper than DFT".
 - I agree that the fact that $\lambda_l^D$ mostly has a small value on  a linear color scale is not evidence that it is not needed, and that, conversely, table 4 confirms its utility.

Weaknesses
 - Quite significant new results at the rebuttal stage (the rebuttal says "incorrectly calculated", it would be good to have supplied more details up front)
 - The rebuttal implies that TorDiff requires 5000 steps.  The reviewer responds that the paper suggests 5-20 steps, and that the time is comparable to RDkit.  The authors do not respond.  This is significant, when we are already agreeing with the authors that a "reasonable" computational burden is acceptable, and given the lack of response, we may assume that this is a valid baseline that has not been compared.

**Justification For Why Not Higher Score:**

This is a good paper on a little-explored aspect of a well-explored problem.  It has the potential to be significant within its domain (molecular machine learning), but it does not present the type of advance that should be disseminated to the wide ML audience.

**Justification For Why Not Lower Score:**

Although it is true that as a poster, this would still reach its core audience, it is valuable to alert practitioners in adjacent subfields to attempts to address new subproblems.

---

### Decision · Program_Chairs · 2024-01-16

Accept (spotlight)